# RETHINKING PARAMETER COUNTING:
# EFFECTIVE DIMENSIONALITY REVISITED

## ABSTRACT

Neural networks appear to have mysterious generalization properties when using parameter counting as a proxy for complexity. Indeed, neural networks often have many more parameters than there are data points, yet still provide good generalization performance. Moreover, when we measure generalization as a function of parameters, we see *double descent* behaviour, where the test error decreases, increases, and then again decreases. We show that many of these properties become understandable when viewed through the lens of *effective dimensionality*, which measures the dimensionality of the parameter space determined by the data. We relate effective dimensionality to posterior contraction in Bayesian deep learning, model selection, width-depth tradeoffs, double descent, and functional diversity in loss surfaces, leading to a richer understanding of the interplay between parameters and functions in deep models. We also show that effective dimensionality compares favourably to alternative norm- and flatness- based generalization measures.

## 1 INTRODUCTION

*Parameter counting* pervades the narrative in modern deep learning. "One of the defining properties of deep learning is that models are chosen to have many more parameters than available training data. In light of this capacity for overfitting, it is remarkable that simple algorithms like SGD reliably return solutions with low test error" (Dziugaite and Roy, 2017). "Despite their massive size, successful deep artificial neural networks can exhibit a remarkably small difference between training and test performance" (Zhang et al., 2017). "Increasing the number of parameters of neural networks can give much better prediction accuracy" (Shazeer et al., 2017). "Scale sensitive complexity measures do not offer an explanation of why neural networks generalize better with over-parametrization" (Neyshabur et al., 2018). "We train GPT-3, an autoregressive language model with 175 billion parameters, $10\times$ more than any previous non-sparse language model" (Brown et al., 2020). The number of model parameters explicitly appears in many modern generalization measures, such as in Equations $20, 51, 52, 56, 57, 59$, and $60$ of the recent study by Jiang et al. (2020). Phenomena such as *double descent* are a consequence of parameter counting. Parameter counting even permeates our language, with expressions such as *over-parametrization* for more parameters than data points.

But parameter counting can be a poor description of model complexity, model flexibility, and inductive biases. One can easily construct degenerate cases, such as predictions being generated by a sum of parameters, where the number of parameters is divorced from the statistical properties of the model. When reasoning about generalization, *over-parametrization* is besides the point: what matters is how the parameters combine with the functional form of the model.

Indeed, the practical success of convolutional neural networks (CNNs) for image recognition tasks is almost entirely about the inductive biases of convolutional filters, depth, and sparsity, for extracting local similarities and hierarchical representations, rather than flexibility (LeCun et al., 1989; Szegedy et al., 2015). Convolutional neural networks have far fewer parameters than fully connected networks, yet can provide much better generalization. Moreover, width can provide flexibility, but it is *depth* that has made neural networks distinctive in their generalization abilities.

In this paper, we gain a number of insights into modern neural networks through the lens of *effective dimensionality*, in place of simple parameter counting. Effective dimensionality, defined by the eigenspectrum of the Hessian on the training loss (equation 2, Section 2), was used by MacKay (1992a) to measure how many directions in the parameter space had been determined in a Bayesian

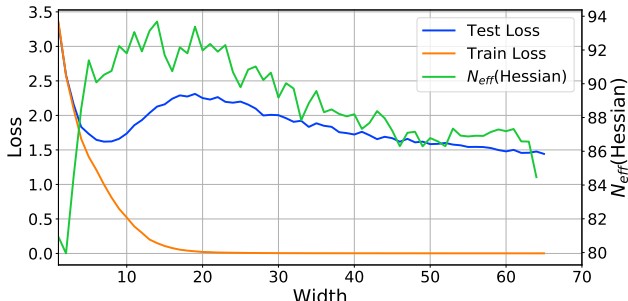

Figure 1: **A resolution of double descent**. We replicate the *double descent* behaviour of deep neural networks using a ResNet18 (He et al., 2016) on CIFAR-100, where train loss decreases to zero with sufficiently wide model while test loss decreases, then increases, and then decreases again.

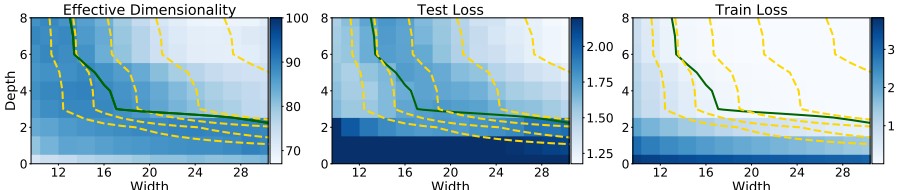

Figure 2: **Left:** Effective dimensionality as a function of model width and depth for a CNN on CIFAR-100. **Center:** Test loss as a function of model width and depth. **Right:** Train loss as a function of model width and depth. Yellow level curves represent equal parameter counts ($1e5$, $2e5$, $4e5$, $1.6e6$). The green curve separates models with near-zero training loss. Effective dimensionality serves as a good proxy for generalization for models with low train loss.

neural network. There is immense value in revisiting effective dimensionality in the context of modern deep learning. We demonstrate that effective dimensionality can be used to explain phenomena such as double descent and width-depth trade-offs in architecture specification (Section 5). We also show that effective dimensionality provides a straightforward, scalable, and promising metric for generalization in modern deep learning, comparing to PAC-Bayes flatness and path-norm measures, two of the most successful measures in the recent study by Jiang et al. (2020) (Section 6). We additionally show how effective dimension can explain why subspace compression methods for neural networks (Izmailov et al., 2019; Li et al., 2018) can be so effective in practice, demonstrating *function-space* homogeneity as we move in directions given by eigenvectors corresponding to the smallest eigenvalues of the Hessian (Section 4). We connect this finding with Bayesian Occam factors and minimum description length frameworks, providing an interpretation of effective dimensionality as model compression (Section 4.3). Moreover, we show that despite a seeming lack of determination in parameter space, a neural network can be relatively well-determined in function space (Section 4).

Consider Figure 1, where we see that once a model has achieved low training loss, the effective dimensionality, computed from training data alone, replicates double descent behaviour for neural networks. Models that are wider have both lower effective dimensionality and better generalization. Alternatively, in Figure 2 we see that width and depth determine effective dimensionality in different ways, though both are related to numbers of parameters. Remarkably, for models with low training loss (above the green partition), the effective dimensionality closely tracks generalization performance for each combination of width and depth. We also see that wide but shallow models overfit, while depth helps provide lower effective dimensionality. When two models have the same training loss they can be viewed as providing a compression of the training data at the same fidelity, in which case the model with the lower effective dimensionality, and thus provides the better compression (Section 4.3) — capturing more regularities — will tend to generalize better. In particular, effective dimension should be used to compare models with similarly low values of training loss. In this regime, we see that ED closely tracks generalization for both double descent and width-depth trade-offs.

## 2 POSTERIOR CONTRACTION, EFFECTIVE DIMENSION, AND THE HESSIAN

We consider a model, typically a neural network, $f(x; \boldsymbol{\theta})$, with inputs $x$ and parameters $\boldsymbol{\theta} \in \mathbb{R}^k$. We define the Hessian as the $k \times k$ matrix of second derivatives of the loss, $\mathcal{H}_{\boldsymbol{\theta}} = -\nabla\nabla_{\boldsymbol{\theta}}\mathcal{L}(\boldsymbol{\theta}, \mathcal{D})$, where $\mathcal{D}$ is the training data. To begin, we describe posterior contraction, effective dimensionality, and connections to the Hessian.

### 2.1 THE HESSIAN AND THE POSTERIOR DISTRIBUTION

We begin by providing a simple example explaining the relationship between the posterior distribution over the model's parameter, the amount of *posterior contraction* from the prior, and the Hessian of the negative log posterior. Figure 3 shows the prior and posterior distribution for a Bayesian linear regression model with a single parameter, with predictions generated by parameters drawn from these distributions. As expected, we see that the variance of the posterior distribution is significantly reduced from that of the prior; we call the difference of the variance between the posterior and the prior the *posterior contraction* of the model. More specifically, as shown in Figure 3, the arrival of data increases the curvature of the loss (negative log

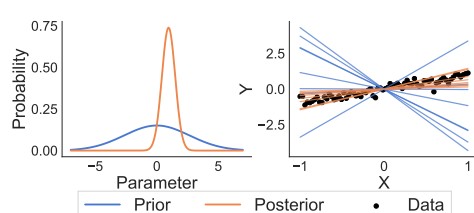

Figure 3: **Left:** A comparison of prior and posterior distributions in a linear regression setting, demonstrating the decrease in variance referred to as *posterior contraction*. **Right:** Functions sampled from the prior and posterior distributions.

posterior) at the optimum. This increase in curvature of the loss that accompanies certainty about the parameters leads to an increase in the eigenvalues of the Hessian of the Growth in eigenvalues of the Hessian of the loss corresponds to increased certainty about parameters.

### 2.2 POSTERIOR CONTRACTION AND EFFECTIVE DIMENSIONALITY

When combined with the functional form of a model, a distribution over parameters $p(\boldsymbol{\theta})$ induces a distribution over functions $p(f(x; \boldsymbol{\theta}))$. The parameters are of little direct interest — what matters for generalization is the distribution over functions (e.g. the right panel of Figure 3). As parameter distributions concentrate around specific values we expect to generate less diverse functions, the behavior seen in Figure 3. We show in Appendix E that we can describe posterior contraction in Bayesian linear regression, $y \sim \mathcal{N}(\boldsymbol{\Phi}\boldsymbol{\beta}, \sigma^2 I)$, with isotropic Gaussian prior, $\boldsymbol{\beta} \sim \mathcal{N}(0, \alpha^2 I_k)$, as

$$\Delta_{post}(\boldsymbol{\Phi}) = \alpha^2 \sum_{i=1}^{N} \frac{\lambda_i}{\lambda_i + \alpha^{-2}}, \tag{1}$$

where $\lambda_i$ are the eigenvalues of $\boldsymbol{\Phi}^\top\boldsymbol{\Phi}/\alpha^2$, the Hessian of the log likelihood, which is dependent on both the model and the training data. We refer to the summation in equation 1 as the *effective dimensionality* of $\boldsymbol{\Phi}^\top\boldsymbol{\Phi}/\alpha^2$. We generalize equation 1, defining the effective dimensionality of a symmetric matrix $\boldsymbol{A} \in \mathbb{R}^{k \times k}$ as

$$N_{eff}(\boldsymbol{A}, z) = \sum_{i=1}^{k} \frac{\lambda_i}{\lambda_i + z}, \tag{2}$$

in which $\lambda_i$ are the eigenvalues of $\boldsymbol{A}$ and $z > 0$ is a regularization constant MacKay (1992a).[1]

Typically as neural networks are trained we observe a gap in the eigenspectrum of the Hessian of the loss (Sagun et al., 2017); a small number of eigenvalues become large while the rest are near zero. In computing effective dimensionality, eigenvalues much larger than $z$ contribute a value of approximately one to the summation, and eigenvalues much smaller than $z$ contribute a value of approximately zero.

---

[1]In discussing model generalization, we use *effective dimensionality* as short form for the effective dimensionality of the Hessian of the loss of a trained model.

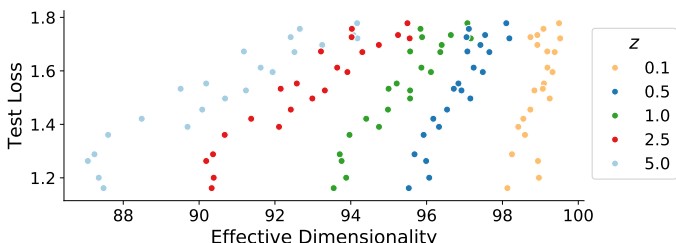

Figure 4: We see a similarly strong relative relationship between effective dimensionality and test loss over a wide range of regularization parameters $z$. Each point is the effective dimensionality computed for a regularization parameter (varying colors) and a particular width and depth, as in Figure 2. We use only models with low training loss (above the green partition in Figure 2) for this plot.

Therefore, the effective dimensionality explains the number of parameters that have been determined by the data, which corresponds to the number of parameters the model is using to make predictions. In comparing models of the same parameterization that achieve low loss on the training data, we expect models with *lower* effective dimensionality to generalize better — which is empirically verified in Figures 1 and 2. The intuition built using Figure 3 carries through to this approximation: as the eigenvalues of the Hessian increase, the eigenvalues of the covariance matrix in our approximation to the posterior distribution shrink, further indicating contraction around the MAP estimate.

**Practical Computations** For large neural networks computing the eigenvalues and eigenvectors of the Hessian of the loss is nontrivial. We estimate effective dimensionality by computing the dominant eigenvalues using the Lanczos algorithm implemented in GPyTorch, since many of the eigenvalues are typically close to zero and do not significantly contribute to the estimate (Gardner et al., 2018). Hessian vector products as implemented in PyTorch take *three* backward passes; In our experiments, we compute 100 Hessian vector products to produce 100 eigenvalues, so that the cost is about $1.5\times$ the cost of standard training [2]. As a heuristic, one can set $z$ using the connection with the prior variance $\alpha^2$ and $\ell_2$ regularization, or to measure the number of relatively large eigenvalues. We use a value of $z = 1$ in equation 2 for all experiments, and show in Figure 4 that effective dimensionality for model comparison is highly robust to different values of $z$ over the range of networks with near zero training loss in Figure 2. For neural networks, the Hessian can have negative eigenvalues (e.g., Sagun et al., 2017; Ghorbani et al., 2019); however, these negative eigenvalues are in practice extremely small in magnitude compared to the positive ones, and do not practically impact the computations of effective dimensionality.

The Hessian (and its effective dimensionality) is not invariant to re-parameterizations (e.g. ReLU rescaling and batch normalization) (MacKay, 2003, Chapter 27). For this reason we assume a fixed parameterization, as is the case in practice, and compare between models of the same parameterization.

## 3 RELATED WORK

MacKay (1992a) used effective dimensionality to measure posterior contraction in Bayesian neural networks. Effective dimensionality has also been used for measuring generalization error of kernel methods (Caponnetto and Vito, 2007).

Various connections between flatness and generalization have been explored via Occam factors (MacKay, 2003; Smith and Le, 2018) and minimum description length (Hinton and Van Camp, 1993; Achille and Soatto, 2018). Nakkiran et al. (2020) found generalization gains as neural networks become overparameterized, showing the *double descent* phenomenon (e.g., Belkin et al., 2019a; Nakkiran et al., 2020) that occurs as the width increases in residual and convolutional neural networks.

Flatness has also be considered in the PAC-Bayes literature (e.g., Dziugaite and Roy, 2017; Jiang et al., 2020), with Jiang et al. (2020) showing that PAC-Bayesian measures of flatness, in the sense of

---

[2]In Appendix I we provide an example of the insensitivity of effective dimensionality to the number of eigenvalues used.

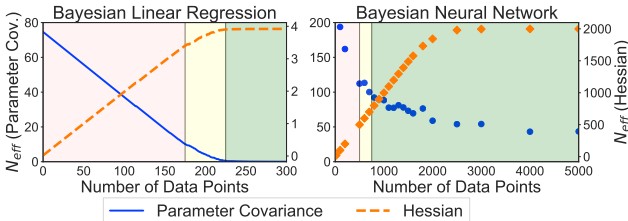

Figure 5: The effective dimensionality of the posterior covariance over parameters and the function-space posterior covariance. Red indicates the over-parameterized setting, yellow the critical regime with $k \approx n$, and green the under-parameterized regime. In both models we see the expected increase in effective dimensionality in parameter space and decrease in effective dimensionality of the Hessian.

insensitivity to random perturbations, perform well relative to many generalization bounds. Zhou et al. (2018) used PAC-Bayesian compression arguments to construct non-vacuous bounds.

Our work shows that effective dimensionality can shed light on a number of phenomena in modern deep learning, including double descent, width-depth trade-offs, and Bayesian subspace inference, while providing a straightforward and compelling generalization metric, relative to several of the highest performing metrics in Jiang et al. (2020). We provide an extended discussion of historical perspectives in Appendix D.

## 4 POSTERIOR CONTRACTION AND FUNCTION-SPACE HOMOGENEITY

We demonstrate how the effective dimensionality of both the posterior parameter covariance and the Hessian of the loss provides insights into how a model adapts to data during training. Additionally we show that in directions in which the posterior has not contracted, neural networks are insensitive to parameter perturbations. In other words, we show that although there are many flat directions in a neural network posterior, the functions obtained by moving along those directions are remarkably homogeneous. Finally we relate insensitivity to parameter perturbations to Occam factors, connecting effective dimensionality to data compression. These findings help explain the surprising success of subspace inference methods for capturing functional variability in low dimensional subspaces.

### 4.1 POSTERIOR CONTRACTION OF BAYESIAN MODELS

**Theorem 4.1** (Posterior Contraction in Bayesian Linear Models). *Let $\boldsymbol{\Phi} = \boldsymbol{\Phi}(x) \in \mathbb{R}^{n \times k}$ be a feature map of $n$ data observations, $x$, with $k$ features, such that $n < k$ and assign isotropic prior $\boldsymbol{\beta} \sim \mathcal{N}(0_k, \alpha^2 I_k)$ for parameters $\boldsymbol{\beta} \in \mathbb{R}^k$. Assuming a model of the form $y \sim \mathcal{N}(\boldsymbol{\Phi}\boldsymbol{\beta}, \sigma^2 I_n)$ the posterior distribution of $\boldsymbol{\beta}$ has a $k - n$ directional subspace in which the variance is identical to the prior variance.*

Theorem 4.1, along with an equivalent result for generalized linear models, is proven in Appendix F.1. Theorem 4.1 demonstrates why *parameter counting* often makes little sense: for a fixed data set of size $n$, only $\min(n, k)$ parameters can be determined, leaving many dimensions in which the posterior is unchanged from the prior when $k \gg n$.

While much effort has been spent grappling with the challenges of high dimensional parameter spaces for Bayesian neural networks, the practical existence of subspaces where the posterior variance has not collapsed from the prior suggests that both computational and approximation gains can be made from ignoring directions in which the posterior variance is unchanged from the prior. This observation helps explain the success of subspace based techniques (e.g., Izmailov et al., 2019) that marginalize the loss in a lower dimensional space. For Bayesian linear models, the effective dimensionality of the parameter covariance is the inverse of the Hessian, and as the effective dimensionality of the parameter covariance decreases, the effective dimensionality of the Hessian increases. Supporting this argument, in Figure 5 we demonstrate for both Bayesian linear models and Bayesian neural networks that as the number of data points grows, the effective dimensionality of the posterior covariance decreases while the effective dimensionality of the Hessian increases until both are limited by the number of parameters. The details of the experimental setup for Figure 5 are given in Appendix B.

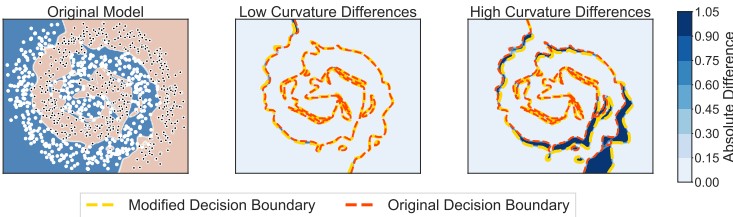

Figure 6: Swiss roll data. **Left:** Adam trained feed-forward, fully connected classifier. **Center:** Differences in original and perturbed classifier when parameters are perturbed in low curvature degenerate directions. **Right:** Differences in the original and perturbed classifier when parameters are perturbed in high curvature directions. **Note** the perturbation in the center plot is approximately 100 *times* the size of that of the plot on the right!

## 4.2 Function-Space Homogeneity

The previous section has demonstrated that parameters can be *undetermined* by the data; we now show that the function induced by the model is *unchanged* in these directions.

**Theorem 4.2** (Function-Space Homogeneity in Linear Models). *Let $\mathbf{\Phi} = \mathbf{\Phi}(x) \in \mathbb{R}^{n \times k}$ be a feature map of $n$ data observations, $x$, with $n < k$, and assign isotropic prior $\boldsymbol{\beta} \sim \mathcal{N}(0_k, \alpha^2 I_k)$ for parameters $\boldsymbol{\beta} \in \mathbb{R}^k$. The minimal eigenvectors of the Hessian define a $k - n$ dimensional subspace in which parameters can be perturbed without changing the training predictions in function space.*

The proof (along with an extension to generalized linear models) is given in Appendix F.2. Although there may be large regions in parameter-space that lead to low-loss models, many of these models may be homogeneous in function space. We can interpret Theorem 4.2 in terms of the eigenvectors of the Hessian indicating which directions in parameter space have and have not been determined by the data. The dominant eigenvectors of the Hessian (those with the largest eigenvalues) correspond to the directions in which the parameters have been determined from the data and the posterior has contracted significantly from the prior. The minimal eigenvectors (those with the smallest eigenvalues) correspond to the directions in parameter space in which the data has not determined the parameters.

Figure 6 demonstrates that the results of Theorem 4.2 apply to neural networks as well, with large perturbations to the parameters in undetermined directions having little effect on the classification boundary. As parameters are perturbed in directions that have not been determined by the data (minimal eigenvectors of the Hessian), the functional form remains identical to the MAP estimate. Perturbations in determined directions (dominant eigenvectors of the Hessian) yield models that perform poorly on the training data and significantly deviate from the MAP estimate on the test set. The homogeneity found by taking perturbations in undetermined parameter directions shows that even in cases with a lack of determination in parameter space, the *functional form* of the model is well specified by the data. Further results regarding function space homogeneity and the connection to loss surface structure, including the projection of the loss surface of the classifier shown in Figure 6 onto random, low curvature, and high curvature bases, are given in Appendix A.

## 4.3 Effective Dimensionality as Compression

In Section 4.2 we show that effective dimensionality relates to the number of parameter directions in which the functional form of the model is sensitive to perturbations, and that in the low curvature directions given by the Hessian eigenvectors corresponding to the smallest eigenvalues the model outputs are largely unchanged by perturbations to the parameters. The presence of these degenerate directions suggests that we can disregard high dimensional subspaces that contain little information about the model, for compression into a smaller subspace containing only the most important parameter directions given by the eigenvalues of the Hessian. This observation helps explain the practical success of such subspace approaches (Izmailov et al., 2019; Li et al., 2018). In short, if two models have similar and low training loss, then the model with lower effective dimension will be providing a better compression of the training data at a similar level of fidelity, and will therefore tend to generalize better. On the other hand, if models have different training loss, or high training loss,

then it becomes harder to user effective dimensionality to predict generalization, as the models may not have learned much from the data, and therefore the level of compression becomes less relevant.

We can also understand the compression of the data provided by a model in terms of minimum description length, by examining the *Occam factor* of the model evidence (MacKay, 2003, Chapter 28). For model $\mathcal{M}$ with parameters $\theta$, we find the Occam factor in decomposing the evidence as,

$$p(\mathcal{D}|\mathcal{M}_i) \approx \underbrace{p(\mathcal{D}|\theta_{MP}, \mathcal{M})}_{} \times \underbrace{p(\theta_{MP}|\mathcal{M})\det^{-\frac{1}{2}}(\mathcal{H}_\theta/2\pi)}_{},$$

$$\text{Evidence} \approx \quad \text{Likelihood} \quad \times \quad \text{Occam Factor} \tag{3}$$

in which $\mathcal{H}_\theta$ is the Hessian of the loss, and $\theta_{MP}$ is the MAP estimate of the parameters.

As the eigenvalues of the Hessian decay and the effective dimensionality decreases, the determinant of the Hessian also decreases, causing the Occam factor to increase, and the description length to decrease, providing a better compression. MacKay (1992b) and MacKay (2003) contains a further discussion of the connection between Occam factors and minimum description length.

## 5 DOUBLE DESCENT AND WIDTH-DEPTH TRADE-OFFS

### 5.1 DOUBLE DESCENT

Thus far, we have simply demonstrated that effective dimensionality gives a better understanding of diversity in the context of the space of functions induced by neural networks. The phenomenon of *double descent* of the generalization performance in both linear and deep models has attracted recent attention (Belkin et al., 2019a; Nakkiran et al., 2020); here, we explain double descent as a function of effective dimensionality.

Following the experimental setup of Nakkiran et al. (2020), we train ResNet18s (He et al., 2016) with varying width parameters, reproducing their double descent curve.[3] In Figure 1 we see effective dimensionality tracks remarkably well with generalization for models with low training loss — displaying the double descent curve that is seen in the test error. We emphasize again that the effective dimensionality is computed using solely the training data, supporting the hypothesis that the eigenvalues of the Hessian matrix can provide a good proxy for generalization performance. In Appendix H, we show similar findings for linear models and small neural networks.

In short, double descent is an artifact of overfitting. As the dimensionality of the parameter space continues to increase past the point where the corresponding models achieve zero training error, flat regions of the loss occupy a greatly increasing volume (Huang et al., 2019), and are thus more easily discoverable by optimization procedures such as SGD. These solutions have lower effective dimensionality, and thus provide better compressions of the data, as in Section 4.3, and therefore better generalization. In concurrent work, Wilson and Izmailov (2020) show that exhaustive Bayesian model averaging over multiple modes eliminates double descent.

In addition to test loss, we also demonstrate that effective dimensionality tracks double descent in *test error* in Figure A.15 (Appendix). Double descent is clearly present for test loss on CIFAR-100, but not test error. To produce double descent for test error, we follow the setup in Nakkiran et al. (2020) and introduce $20\%$ label corruption, which increases the chance of overfitting.

### 5.2 NETWORKS OF VARYING WIDTH AND DEPTH

Double descent experiments typically only consider increases in width. However, it is *depth* which has endowed neural networks with distinctive generalization properties. In Figure 2, we consider varying both the width and depth of a convolutional neural network on the CIFAR-100 dataset. In the region of near-zero training loss, separated by the green curve, we see effective dimensionality closely matches generalization performance. Moreover, wide but shallow models tend to overfit, providing low training loss, but high effective dimensionality and test loss. On the other hand, deeper models have lower test loss and lower effective dimensionality, showing that depth enables a better compression of the data.

---

[3]See Appendix G for training details.

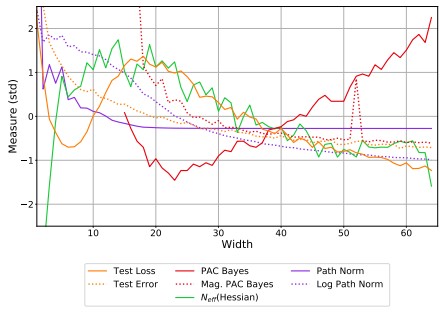
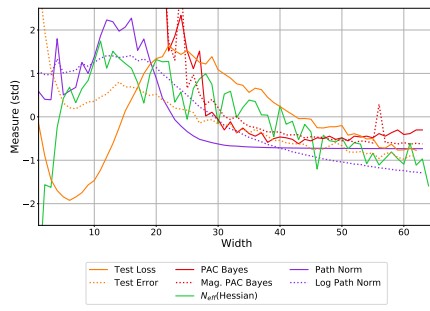

(a) *Double Descent, No Label Noise*       (b) *Double Descent, 20% Label Noise*

Figure 7: Comparing effective dimensionality as a generalization measure to the path-norm, log path-norm, PAC-Bayes, and magnitude aware PAC-Bayes flatness measures from Jiang et al. (2020) for double descent (a) without and (b) with label noise. For models with low train loss, effective dimensionality most closely follows both test error and test loss.

# 6   EFFECTIVE DIMENSIONALITY AS A GENERALIZATION MEASURE

We have shown, for the first time, that a generalization measure is able to track and explain double descent and width-depth trade-offs in modern deep networks. We now demonstrate that effective dimensionality is a strong generalization measure by itself. For comparison, we choose path-norm and PAC-Bayesian based sharpness measures, due to their good performance in prior work (Jiang et al., 2020; Keskar et al., 2017; Neyshabur et al., 2017). We compute the norms as in Jiang et al. (2020) for a neural network function $f(x; \boldsymbol{\theta})$ with input $x$ and weights $\boldsymbol{\theta}$.

The path-norm describes the norm of all paths within a network between input and output nodes, measuring the complexity of traversing the network. The standard PAC-Bayesian flatness measure aims to measure the overall flatness of the solution by determining the size of the largest perturbation that can be made to the parameters such that the expected increase in training error remains low. We additionally compare to the magnitude aware PAC-Bayesian flatness measure, which is similar to the standard PAC-Bayesian flatness measure, except that the perturbations to parameters are scaled by the magnitude of the weights in the network. Computation details are given in Appendix C.

We extend the results of Figure 1 in Figure 7 to include the path-norm and PAC-Bayesian flatness measures. We display test loss, test error, and generalization measures, standardized by subtracting the sample mean and dividing by the sample standard deviation. We extend the results of Figure 2 in the same fashion as Figure 7 in the Appendix in Figure A.13. We additionally show correlations with generalization in Tables 1 and 2 in the Appendix.

There are several key take-aways: (1) effective dimension overall provides a better proxy for test loss and test error than the PAC-Bayes or path-norm measures for double descent and width-depth trade-offs; (2) not all flatness-based generalization measures are equivalent, and in fact different flatness measures can provide wildly different behaviour; (3) effective dimension is more stable than the other measures, with relatively consistent behaviour across qualitatively similar datasets and models; (4) effective dimension is *interpretable*, providing a clear connection with function-space and an explanation for *why* a model should generalize, in addition to an association with generalization.

Not all flatness-based generalization measures are equivalent. In contrast to the PAC-Bayesian flatness measure considered in Jiang et al. (2020), effective dimension computes the number of directions in parameter space that are flat, as determined by the curvature of the loss surface. Combined with our observations about function-space homogeneity in Section 4.2, we see that effective dimension provides a proxy for model compression and generalization in addition to providing correlation with performance. By contrast, the PAC-Bayes flatness measure considers the size of the basin surrounding an optimum, and is highly sensitive to the sharpest direction.

# 7 CONCLUSION

We have shown how the effective dimensionality of the Hessian of the loss can be used to gain insight into a range of phenomena, including double descent, posterior contraction, loss surface structure, subspace inference, and function-space diversity of models. As we have seen, simple parameter counting can be a misleading proxy for model complexity and generalization performance; models with many parameters combined with a particular functional form can give rise to simple explanations for data. Indeed, we have seen how depth and width have different effects on generalization performance, regardless of the total number of parameters in the model. In all cases, effective dimensionality tracks generalization performance for models with comparable training loss. Moving forward, we hope our work will help inspire a continued effort to capture the nuanced interplay between the statistical properties of parameter space and function space in understanding generalization behaviour, and highlight effective dimension as a relatively simple and robust generalization measure in modern deep learning.

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

## A  LOSS SURFACES AND FUNCTION SPACE REPRESENTATIONS

Recent works have discussed the desirability of finding solutions corresponding to *flat* optima in the loss surface, arguing that such parameter settings lead to better generalization (Izmailov et al., 2018; Keskar et al., 2017). There are multiple notions of flatness in loss surfaces, relating to both the volume of the basin in which the solution resides and the rate of increase in loss as one moves away from the found solution. As both definitions correspond to low curvature in the loss surface, it is standard to use the Hessian of the loss to examine structure in the loss surface Madras et al. (2019); Keskar et al. (2017).

The effective dimensionality of the Hessian of the loss indicates the number of parameters that have been determined by the data. In highly over-parameterized models we hypothesize that the effective dimensionality is substantially less than the number of parameters, i.e. $N_{eff}(\mathcal{H}_{\boldsymbol{\theta}}, \alpha) \ll p$, since we should be unable to determine many more parameters than we have data observations.

Recall from Section 2.2 the large eigenvalues of the Hessian have eigenvectors corresponding to directions in which parameters are determined. equation 2 dictates that low effective dimensionality (in comparison to the total number of parameters) would imply that there are many directions in which parameters are not determined, and the Hessian has eigenvalues that are near zero, meaning that in many directions the loss surface is constant. We refer to directions in parameter space that have not been determined as *degenerate* for two reasons: (1) degenerate directions in parameter space provide minimal structure in the loss surface, shown in Section A.1; (2) parameter perturbations in degenerate directions do not provide diversity in the function-space representation of the model, shown in Section A.2. We refer to the directions in which parameters have been determined, directions of high curvature, as *determined*.

To empirically test our hypotheses regarding degenerate directions in loss surfaces and function space diversity, we train a neural network classifier on $1000$ points generated from the two-dimensional Swiss roll data, with a similar setup to Huang et al. (2019), using Adam with a learning rate of $0.01$ Kingma and Ba (2015). The network is fully connected, consisting of 5 hidden layers each 20 units wide (plus a bias term), and uses ELU activations with a total of 2181 parameters. We choose a small model with two-dimensional inputs so that we can both tractably compute all the eigenvectors and eigenvalues of the Hessian and visualize the functional form of the model.

### A.1  LOSS SURFACES AS DETERMINED BY THE HESSIAN

To examine the loss surface more closely, we visualize low dimensional projections. To create the visualizations, we first define a basis given by a set of vectors, then choose a two random vectors, $u$ and $\widetilde{v}$, within the span of the basis. We use Gram-Schmidt to orthogonalize $\widetilde{v}$ with respect to $u$, ultimately giving $u$ and $v$ with $u \perp v$. We then compute the loss at parameter settings $\boldsymbol{\theta}$ on a grid surrounding the optimal parameter set, $\boldsymbol{\theta}^*$, which are given by

$$\boldsymbol{\theta} \leftarrow \boldsymbol{\theta}^* + \alpha u + \boldsymbol{\beta} v \qquad (4)$$

for various $\alpha$ and $\boldsymbol{\beta}$ values such that all points on the grid are evaluated.

By selecting the basis in which $u$ and $v$ are defined we can specifically examine the loss in determined and degenerate directions. Figure A.8 shows that in determined directions, the optimum appears

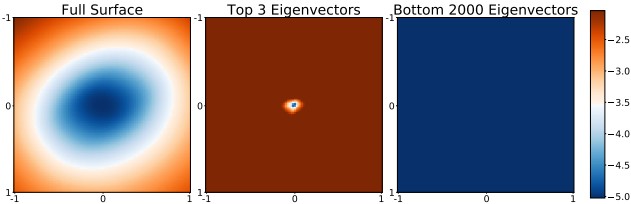

Figure A.8: **Left:** A random projection of the loss surface. **Center:** A projection of the loss surface in the top 3 directions in which parameters have been determined. **Right:** A projection of the loss surface in the 2000 (out of 2181) directions in which parameters have been determined the least. The rightmost plot shows that in degenerate parameter directions the loss is constant.

extremely sharp. Conversely, in all but the most determined directions, the loss surface loses all structure and appears constant. Even in degenerate directions, if we deviate from the optimum far enough the loss will eventually become large. However to observe this increase in loss requires perturbations to the parameters that are significantly larger in norm than $\boldsymbol{\theta}^*$.

### A.2 DEGENERATE PARAMETERS LEAD TO HOMOGENEOUS MODELS

In this section we show that degenerate parameter directions do not contain diverse models. This result is not at odds with the notion that flat regions in the loss surface can lead to diverse but high performing models. Rather, we find that there is a subspace in which the loss is constant and one cannot find model diversity, noting that this subspace is distinct from those employed by works such as Izmailov et al. (2019) and Huang et al. (2019). This finding leads to an interpretation of effective dimensionality as *model compression*, since the undetermined directions do not contain additional functional information.

We wish to examine the functional form of models obtained by perturbing the parameters found through training, $\boldsymbol{\theta}^*$. Perturbed parameters are computed as

$$\boldsymbol{\theta} \leftarrow \boldsymbol{\theta}^* + s\frac{Bv}{||Bv||_2} \tag{5}$$

where $B \in \mathbb{R}^{k \times d}$ is a $d$ dimensional basis in which we wish to perturb $\boldsymbol{\theta}^*$, and $v \sim \mathcal{N}(0, I_d)$, giving $Bv$ as a random vector from within the span of some specified basis (i.e. the dominant or minimal eigenvectors). The value $s$ is chosen to determine the scale of the perturbation, i.e. the length of the random vector by which the parameters are perturbed.

Experimentally, we find that in a region near the optimal parameters $\boldsymbol{\theta}^*$, i.e. $s \leq ||\boldsymbol{\theta}^*||_2/2$ the function-space diversity of the model is contained within the subspace of determined directions. While the degenerate directions contain wide ranges of parameter settings with low loss, the models are equivalent in function space.

Figure 6 shows the trained classifier and the differences in function-space between the trained classifier and those generated from parameter perturbations. We compare perturbations of size $||\boldsymbol{\theta}^*||_2/2 \approx 10$ in the direction of the 500 minimal eigenvectors and perturbations of size 0.1 in the directions of the 3 maximum eigenvectors. A perturbation from the trained parameters in the directions of low curvature (center plot in Figure 6) still leads to a classifier that labels all points identically. A perturbation roughly 100 times smaller the size in directions in which parameters have been determined leads to a substantial change in the decision boundary of the classifier.

However, the change in the decision boundary resulting from perturbations in determined directions is not necessarily desirable. One need not perturb parameters in either determined or degenerate directions to perform a downstream task such as ensembling. Here, we are showcasing the degeneracy of the subspace of parameter directions that have not been determined by the data. This result highlights that despite having many parameters the network could be described by a relatively low dimensional subspace.

[Finally, Figure A.9 demonstrates that the degenerate directions in parameter space lead to models that are homogeneous in function space on both training and testing data. As increas-

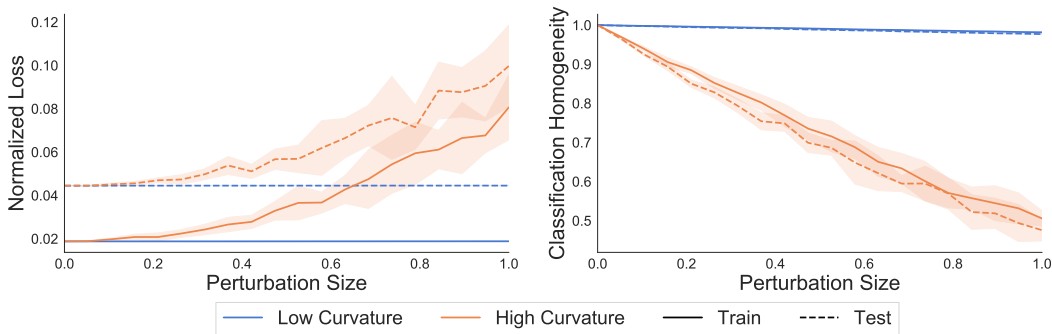

Figure A.9: **[Left: Loss, normalized by dataset size, on both train and test sets as perturbations are made in high curvature directions and degenerate directions. Right: Classification homogeneity, the fraction of data points classified the same as the unperturbed model, as perturbations are made in both high curvature and degenerate directions.]**

**ingly large perturbations are made in degenerate parameter directions, we still classify more than $99\%$ of both training and testing points the same as the unperturbed classifier.]**

## B  EFFECTIVE DIMENSIONALITY OF THE POSTERIOR COVARIANCE

**Linear Models:**  We construct $\mathbf{\Phi}(x)$ with each row as an instance of a 200 dimensional feature vector consisting of sinusoidal terms for each of 500 observations: $\mathbf{\Phi}(x) = [\cos(\pi x), \sin(\pi x), \cos(2\pi x), \sin(2\pi x), \dots]$. We assign the coefficient vector $\boldsymbol{\beta}$ a prior $\boldsymbol{\beta} \sim \mathcal{N}(0, I)$, and draw ground truth parameters $\boldsymbol{\beta}^*$ from this distribution. The model takes the form $\boldsymbol{\beta} \sim \mathcal{N}(0, I)$ and $y \sim \mathcal{N}\left(\mathbf{\Phi}\boldsymbol{\beta}, \sigma^2 I\right)$. We randomly add data points one at a time while re-computing the posterior covariance. We compute the effective dimensionality, $N_{eff}\left(\Sigma_{\boldsymbol{\beta}|\mathcal{D},\sigma}, \alpha\right)$, where $\Sigma_{\boldsymbol{\beta}|\mathcal{D},\sigma}$ is the posterior covariance of $\boldsymbol{\beta}$.[4]

**Small Bayesian Neural Networks**  For the experiments with small Bayesian neural networks, we generate a nonlinear function of the form, $y = w_1 x + w_2 x^2 + w_3 x^3 + (0.5 + x^2)^2 + \sin(4x^2) + \epsilon$, with $w_i \sim \mathcal{N}(0, I)$ and $\epsilon \sim \mathcal{N}(0, 0.05^2)$, and de-mean and standardize the inputs.[5] We then construct a Bayesian neural network with two hidden layers each with 20 units, no biases, and $tanh$ activations, placing independent Gaussian priors with variance 1 on all model parameters. We then run the No-U-Turn sampler (Hoffman and Gelman, 2014) for 2000 burn-in iterations before saving the final 2000 samples from the approximated posterior distribution. Using these samples, we compute the effective dimensionality of the sample posterior covariance, $\text{Cov}_{p(\boldsymbol{\theta}|\mathcal{D})}(\boldsymbol{\theta})$, and Hessian of the loss at the MAP estimate in Figure 5. The trends of effective dimensionality for Bayesian neural networks are aligned with Bayesian linear regression, with the effective dimensionality of the Hessian (corresponding to function space) increasing while the effective dimensionality of the parameter space decreases.

## C  COMPUTATION OF THE PAC BAYES NORMS

The path-norm is the square root of the sum of the outputs produced by a forwards pass on an input of all ones,

$$\mu_{\text{path-norm}}(f) = \left(\sum f(\mathbf{1}; \boldsymbol{\theta}^2)\right)^{1/2}. \tag{6}$$

with the parameters $\boldsymbol{\theta}$ squared (Eq. 44 of Jiang et al. (2020)). Through squaring the weights and taking the square root of the output, we form a correspondence between the path-norm and the $\ell_2$ norm of all paths within a network from an input node to an output node (Neyshabur et al., 2017).

---

[4]Here we use $\alpha = 5$, however the results remain qualitatively the same as this parameter changes.

[5]From the Bayesian neural network example in NumPyro (Phan et al., 2019; Bingham et al., 2019): `https://github.com/pyro-ppl/numpyro/blob/master/examples/bnn.py`.

The PAC-Bayesian flatness measure of Jiang et al. (2020), adapted from Dziugaite and Roy (2017) and Keskar et al. (2017), is perturbation-based and computed as

$$\mu_{\text{pac-bayes-sharpness}}(f) = \frac{1}{\sigma^2} \,, \tag{7}$$

where $\sigma$ is the largest value such that

$$\mathbb{E}_{u \sim \mathcal{N}(0, \sigma^2 I)} \left[ \mathcal{L}(\boldsymbol{\theta} + u, \mathcal{D}) \right] \leq \mathcal{L}(\boldsymbol{\theta}, \mathcal{D}) + 0.1 \,. \tag{8}$$

In equation 8, $\mathcal{L}(\boldsymbol{\theta}, \mathcal{D})$ is the prediction *error* on the training dataset of the network with weights $\boldsymbol{\theta}$ as computed on data set $\mathcal{D}$. This measure corresponds to a bound on parameter perturbations such that increases in training error remain beneath $0.1$ in expectation as in Jiang et al. (2020).

We additionally compare to the magnitude aware PAC-Bayes bound,

$$\mu_{\text{mag-pac-bayes-sharpness}}(f) = \frac{1}{\sigma'^2} \,, \tag{9}$$

where $\sigma'$ is the largest value such that

$$\mathbb{E}_{u \sim \mathcal{N}(0, \sigma'^2 |\boldsymbol{\theta}| + \epsilon)} \left[ \mathcal{L}(\boldsymbol{\theta} + u, \mathcal{D}) \right] \leq \mathcal{L}(\boldsymbol{\theta}, \mathcal{D}) + 0.1 \,. \tag{10}$$

In equation 10 the variance of the perturbation to each parameter is scaled according to the magnitude of that parameter, adding stability by accounting for differences in scales, and, implicitly, the size of the perturbation with the dimension of the parameter space. The value of $\epsilon$ is taken to be $0.001$ as in Jiang et al. (2020) and serves to regularize the distribution, preventing the distribution from collapsing in the presence of weights that are close to $0$.

In general, path-norm acts only on model parameters, and is thus not directly connected to function-space or the shape of the loss. It should therefore be used with particular caution in comparing different architectures. Indeed, we found that path-norm tends to quickly saturate with increases in model size, with no preference between larger models even when these models provide very different generalization performance. We exhaustively considered how path-norm could be made to provide reasonable performance on these problems. For smaller models, the path-norm is orders of magnitude larger than the path-norm computed on larger models. We therefore performed a log transform, which to our knowledge has not been considered before. We surprisingly found the log transform dramatically improves the correlation of path-norm to generalization performance for comparing amongst large convolutional nets and residual nets. However, this modified measure is still only associated with generalization, is highly sensitive to experimental details (e.g. size of models being compared), and does not provide any direct intuition for model comparison given its reliance on parameters alone.

## D  HISTORICAL PERSPECTIVES ON EFFECTIVE DIMENSIONALITY

Here, we provided a detailed description of the history of effective dimensionality and related measures of flatness and generalization. Cleveland (1979) introduced effective dimensionality into the splines literature as a measure of goodness of fit, while Hastie and Tibshirani (1990, Chapter 3) used it to assess generalized additive models. Gull (1989) first applied effective dimensionality in a Bayesian setting for an image reconstruction task, while MacKay (1992a;b) used it to compute posterior contraction in Bayesian neural networks. Moody (1992) argued for the usage of the effective dimensionality as a proxy for generalization error, while Moody (1991) suggested that effective dimensionality could be used for neural network architecture selection. Zhang (2005) and Caponnetto and Vito (2007) studied the generalization abilities of kernel methods in terms of the effective dimensionality. Achille and Soatto (2018) argue that flat minima have low information content (many small magnitude eigenvalues of the Hessian) by connecting PAC-Bayesian approaches to information theoretic arguments, before demonstrating that low information functions learn invariant representations of the data.

Friedman et al. (2001, Chapter 7) use the effective dimensionality (calling it the effective degrees of freedom) to compute the expected generalization gap for regularized linear models. Dobriban and Wager (2018) specifically tied the bias variance decomposition of predictive risk in ridge regression (e.g. the finite sample predictive risk under Gaussian priors) to the effective dimensionality of the

feature matrix, $\boldsymbol{\Phi}^\top\boldsymbol{\Phi}$. Hastie et al. (2019), Muthukumar et al. (2019), Bartlett et al. (2019), Mei and Montanari (2019), and Belkin et al. (2019b) studied risk and generalization in over-parameterized linear models, including under model misspecification. Bartlett et al. (2019) also introduced the concept of effective rank of the feature matrix, which has a similar interpretation to effective dimensionality.

Sagun et al. (2017) found that the eigenvalues of the Hessian increase through training, while Papyan (2018) and Ghorbani et al. (2019) studied the eigenvalues of the Hessian for a range of modern neural networks. Suzuki (2018) produced generalization bounds on neural networks via the effective dimensionality of the covariance of the functions at each hidden layer. Fukumizu et al. (2019) embedded narrow neural networks into wider neural networks and studied the flatness of the resulting minima in terms of their Hessian via a PAC-Bayesian approach.

Moreover, MacKay (2003) and Smith and Le (2018) provide an Occam factor perspective linking flatness and generalization. Related minimum description length perspectives can be found in MacKay (2003) and Hinton and Van Camp (1993). Other works also link flatness and generalization (e.g., Hochreiter and Schmidhuber, 1997; Keskar et al., 2017; Chaudhari et al., 2019; Izmailov et al., 2018), with Izmailov et al. (2018) and Chaudhari et al. (2019) proposing optimization procedures for flat minima.

To the best of our knowledge, Opper et al. (1989), Opper et al. (1990), Bös et al. (1993), and LeCun et al. (1991) introduced the idea that generalization error for neural networks can decrease, increase, and then again decrease with increases in parameters (e.g. the double descent curve) while Belkin et al. (2019a) re-introduced the idea into the modern machine learning community by demonstrating its existence on a wide variety of machine learning problems. Nakkiran et al. (2020) found generalization gains as neural networks become highly overparameterized, showing the *double descent* phenomenon that occurs as the width parameter of both residual and convolutional neural networks is increased.

**[Similar to our work, both Ghorbani et al. (2019) and Yao et al. (2019) use different implementations of the Lanczos algorithm to estimate eigenvalues of the Hessian; however, they focus on either estimating the spectral density (both) or on the trace of the Hessian (only Yao et al. (2019)), with Ghorbani et al. (2019)'s work implemented in Tensorflow and Yao et al. (2019) also implementing Lanczos in PyTorch.]**

## E  MEASURING POSTERIOR CONTRACTION IN BAYESIAN GENERALIZED LINEAR MODELS

We first consider the over-parametrized case, $k > n$:

$$\Delta_{post}(\boldsymbol{\theta}) = tr(Cov_{p(\boldsymbol{\theta})}(\boldsymbol{\theta})) - tr(Cov_{p(\boldsymbol{\theta}|\mathcal{D})}(\boldsymbol{\theta})) = \sum_{i=1}^{k}\alpha^2 - \sum_{i=1}^{n}(\lambda_i + \alpha^{-2})^{-1} + \sum_{i=n+1}^{k}\alpha^2$$

$$= k\alpha^2 - (k-n)\alpha^2 - \sum_{i=1}^{n}(\lambda_i + \alpha^{-2})^{-1} = \sum_{i=1}^{n}\frac{1 - \alpha^2(\lambda_i + \alpha^{-2})}{\lambda_i + \alpha^{-2}}$$

$$= \alpha^2 \sum_{i=1}^{n}\frac{\lambda_i}{\lambda_i + \alpha^{-2}}; \tag{11}$$

where we have used Theorem 4.1 to assess the eigenvalues of the posterior covariance. When $n > k$, we have the simplified setting where the summation becomes to $k$ instead of $n$, giving us that all of the eigenvalues are shifted from their original values to become $\lambda_i + \alpha^{-2}$, and so

$$\Delta_{post.}(\boldsymbol{\theta}) = \alpha^{-2} \sum_{i=1}^{k}\frac{\lambda_i}{\lambda_i + \alpha^{-2}}, \tag{12}$$

where $\lambda_i$ is the $i$th eigenvalue of $\boldsymbol{\Phi}^\top\boldsymbol{\Phi}/\sigma^2$.

### E.1 Contraction in Function Space

We can additionally consider the posterior contraction in function space. For linear models, the posterior covariance on the training data in function space becomes

$$\boldsymbol{\Phi}\Sigma_{\boldsymbol{\beta}|\mathcal{D}}\boldsymbol{\Phi}^\top = \sigma^2\boldsymbol{\Phi}(\boldsymbol{\Phi}^\top\boldsymbol{\Phi} + \frac{\sigma^2}{\alpha^2}I_p)^{-1}\boldsymbol{\Phi}^\top, -x \tag{13}$$

while the prior covariance in function space is given by $\alpha^2\boldsymbol{\Phi}\boldsymbol{\Phi}^\top$. We will make the simplifying assumption that the features are normalized such that $tr(\boldsymbol{\Phi}\boldsymbol{\Phi}^\top) = rank(\boldsymbol{\Phi}\boldsymbol{\Phi}^\top) = r$. Now, we can simplify

$$\Delta_{post}(f) = tr(Cov_{p(f)}(f) - tr(Cov_{p(f|\mathcal{D})}(f)) = \alpha^2 r - \sigma^2\sum_{i=1}^{r}\frac{\lambda_i}{\lambda_i + \sigma^2/\alpha^2}$$

$$= \alpha^2\sum_{i=1}^{r}\frac{\lambda_i + \sigma^2/\alpha^2}{\lambda_i + \sigma^2/\alpha^2} - \sigma^2\sum_{i=1}^{r}\frac{\lambda_i}{\lambda_i + \sigma^2/\alpha^2}$$

$$= (\alpha^2 - \sigma^2)\sum_{i=1}^{r}\frac{\lambda_i}{\lambda_i + \sigma^2/\alpha^2} + \sigma^2\sum_{i=1}^{r}\frac{1}{\lambda_i + \sigma^2/\alpha^2}.$$

Simplifying and recognizing these summations as the effective dimensionalities of $\boldsymbol{\Phi}^\top\boldsymbol{\Phi}$ and $(\boldsymbol{\Phi}^\top\boldsymbol{\Phi})^+$, we get that

$$\Delta_{post}(f) = (\alpha^2 - \sigma^2)N_{eff}(\boldsymbol{\Phi}^\top\boldsymbol{\Phi}, \sigma^2/\alpha^2) + \sigma^2 N_{eff}((\boldsymbol{\Phi}^\top\boldsymbol{\Phi})^+, \alpha^2/\sigma^2) \tag{14}$$

$$= \sigma^2 r + (\alpha^2 - 2\sigma^2)N_{eff}(\boldsymbol{\Phi}^\top\boldsymbol{\Phi}, \sigma^2/\alpha^2),$$

thereby showing that the posterior contraction in function space is explicitly tied to the effective dimensionality of the Gram matrix.

## F  Posterior Contraction and Function-Space Homogeneity Proofs and Additional Theorems

In this section we complete the proofs to Theorems 4.1 and 4.2 and extend the results from linear models to generalized linear models.

### F.1  Proof and Extensions to Theorem 4.1

**Theorem** (Posterior Contraction in Bayesian Linear Models). *Let $\boldsymbol{\Phi} = \boldsymbol{\Phi}(x) \in \mathbb{R}^{n \times k}$ be a feature map of $n$ data observations, $x$, with $n < k$ and assign isotropic prior $\boldsymbol{\beta} \sim \mathcal{N}(0_k, S_0 = \alpha^2 I_k)$ for parameters $\boldsymbol{\beta} \in \mathbb{R}^k$. Assuming a model of the form $y \sim \mathcal{N}(\boldsymbol{\Phi}\boldsymbol{\beta}, \sigma^2 I_n)$ the posterior distribution of $\boldsymbol{\beta}$ has an [k − n] directional subspace in which the variance is identical to the prior variance.*

*Proof.* The posterior distribution of $\boldsymbol{\beta}$ in this case is known and given as

$$\boldsymbol{\beta}|\mathcal{D} \sim \mathcal{N}((\mu|\mathcal{D}), (\Sigma|\mathcal{D}))$$
$$\mu|\mathcal{D} = (\boldsymbol{\Phi}^\top\boldsymbol{\Phi}/\sigma^2 + S_0^{-1})^{-1}\boldsymbol{\Phi}^\top y/\sigma^2 \tag{15}$$
$$\Sigma|\mathcal{D} = (\boldsymbol{\Phi}^\top\boldsymbol{\Phi}/\sigma^2 + S_0^{-1})^{-1}$$

We want to examine the distribution of the eigenvalues of the posterior variance. Let $\boldsymbol{\Phi}^\top\boldsymbol{\Phi}/\sigma^2 = V\lambda_n V^\top$ be the eigendecomposition with eigenvalues $\Lambda = \text{diag}(\gamma_1, \ldots, \gamma_n, 0_{n+1}, \ldots, 0_k)$; $k - n$ of the eigenvalues are 0 since the gram matrix $\boldsymbol{\Phi}^\top\boldsymbol{\Phi}$ is at most rank [n] by construction. Substitution into the posterior variance of $\boldsymbol{\beta}$ yields,

$$(\boldsymbol{\Phi}^\top\boldsymbol{\Phi}/\sigma^2 + S_0^{-1})^{-1} = (V\Lambda V^\top + \alpha^{-2}I_k)^{-1} = V(\Lambda + \alpha^{-2}I_k)^{-1}V^\top$$
$$= V\Gamma V^\top. \tag{16}$$

The eigenvalues of the posterior covariance matrix are given by the entries of $\Gamma$, $\left((\gamma_1 + \alpha^{-2})^{-1}, \ldots, (\gamma_n + \alpha^{-2})^{-1}, \alpha^2, \ldots, \alpha^2\right)$, where there are $k - n$ eigenvalues that retain a value of $\alpha^2$. **[Therefore, the posterior covariance has $k - n$ directions in which the posterior variance is unchanged and $n$ directions in which it has contracted as scaled by the eigenvalues of the gram matrix $\Phi^\top \Phi$.]** $\qquad\square$

Generalized linear models (GLMs) do not necessarily have a closed form posterior distribution. However, Neal and Zhang (2006) give a straightforward argument using the invariance of the likelihood of GLMs to orthogonal linear transformation in order to justify the usage of PCA as a feature selection step. We can adapt their result to show that overparameterized GLMs have a $k - n$ dimensional subspace in which the posterior variance is identical to the prior variance.

**Theorem F.1** (Posterior Contraction in Generalized Linear Models). *We specify a generalized linear model, $E[Y] = g^{-1}(\Phi\beta)$ and $Var(Y) = V(g^{-1}(\Phi\beta))$, where $\Phi \in \mathbb{R}^{n \times k}$ is a feature matrix of $n$ observations and $k$ features and $\beta \in \mathbb{R}^k$ are the model parameters. In the overparameterized setting with isotropic prior on $\beta$, there exists a $k - n$ dimensional subspace in which the posterior variance is identical to the prior variance.*

*Proof.* First note that the likelihood of a GLM takes as argument $\Phi\beta$, thus transformations that leave $\Phi\beta$ unaffected leave the likelihood, and therefore the posterior distribution, unaffected.

Let $R$ be an orthogonal matrix, $R^\top R = RR^\top = I_p$, and $\tilde{\beta} = R\beta \sim N(0, \sigma^2 I)$. If we assign a standard isotropic prior, to $\beta$ then $\tilde{\beta} = R\beta \sim \mathcal{N}(0, \sigma^2 R I_k R^\top = \sigma^2 I_k)$. If we also rotate the feature matrix, $\tilde{\Phi} = \Phi R^\top \in \mathbb{R}^{n \times k}$ so that $\tilde{\Phi}\tilde{\beta} = \Phi R^\top R\beta = \Phi\beta$, showing that the likelihood and posterior remain unchanged under such transformations.

In the overparameterized regime, $k > n$, with linearly independent features we have that $\Phi$ has rank at most $k$, and we can therefore choose $R$ to be a rotation such that $\Phi R$ has exactly $k - n$ columns that are all 0. This defines a $k - n$ dimensional subspace of $\beta \in \mathbb{R}^k$ in which the the likelihood is unchanged. Therefore the posterior remains no different from the prior distribution in this subspace, or in other words, the posterior distribution has not contracted in $k - n$ dimensions. $\qquad\square$

## F.2    FUNCTION-SPACE HOMOGENEITY

**Theorem** (Function-Space Homogeneity in Linear Models). *Let $\Phi = \Phi(x) \in \mathbb{R}^{n \times k}$ be a feature map of $n$ data observations, $x$, with $n < k$ and assign isotropic prior $\beta \sim \mathcal{N}(0_k, S_0 = \alpha^2 I_k)$ for parameters $\beta \in \mathbb{R}^k$. The minimal eigenvectors of the Hessian define a $k - n$ dimensional subspace in which parameters can be perturbed without changing the training predictions in function-space.*

*Proof.* The posterior covariance matrix for the parameters is given by

$$\Sigma_{\beta|\mathcal{D}} = \left(\frac{\Phi^\top \Phi}{\sigma^2} + \alpha^{-2} I_k\right)^{-1},$$

and therefore the Hessian of the log-likelihood is $\left(\frac{\Phi^\top \Phi}{\sigma^2} + \alpha^{-2} I_k\right)$. By the result in Theorem 4.1 there are $k - n$ eigenvectors of the Hessian all with eigenvalue $\alpha^{-2}$. If we have some perturbation to the parameter vector $u$ that resides in the span of these eigenvectors we have

$$\left(\frac{\Phi^\top \Phi}{\sigma^2} + \alpha^{-2} I_k\right) u = \alpha^{-2} u,$$

which implies $u$ is in the nullspace of $\Phi^\top \Phi$. By the properties of gram matrices we have that the nullspace of $\Phi^\top \Phi$ is the same as that of $\Phi$, thus $u$ is also in the nullspace of $\Phi$ Therefore any prediction using perturbed parameters takes the form $\hat{y} = \Phi(\beta + u) = \Phi\beta$, meaning the function-space predictions on training data under such perturbations are unchanged. $\qquad\square$

**Theorem F.2** (Function-Space Homogeneity in Generalized Linear Models). *We specify a generalized linear model, $E[Y] = g^{-1}(\Phi\beta)$, where $\Phi \in \mathbb{R}^{n \times k}$ is a feature matrix of $n$ observations and $k$ features and $\beta \in \mathbb{R}^k$ are the model parameters. In the overparameterized setting with isotropic prior on $\beta$, there exists a $k - n$ dimensional subspace in which parameters can be perturbed without changing the training predictions in function-space or the value of the Hessian.*

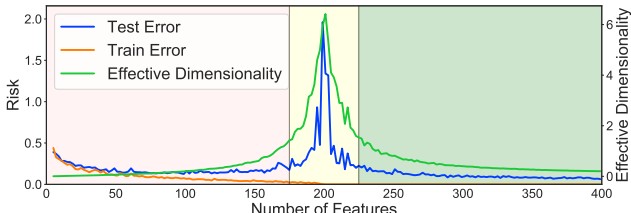

Figure A.10: Demonstration of double descent for linear models with an increasing number of features. We plot the effective dimensionality of the Hessian of the loss. In the regime with near-zero train error, the test error is almost entirely explained by the effective dimensionality, which is computed on the train set alone. The red region corresponds to underparameterized models, yellow to critically parameterized models, and green to overparameterized models.

*Proof.* The Hessian of the log-likelihood for GLMs can be written as a function of the feature map, $\boldsymbol{\Phi}$, and the product of the feature map and the parameters, $\boldsymbol{\Phi}\boldsymbol{\beta}$, i.e. $\boldsymbol{\beta}$ only appears multiplied by the feature map (Nelder and Wedderburn, 1972). We can then write $\mathcal{H}_{\boldsymbol{\beta}} = f(\boldsymbol{\Phi}\boldsymbol{\beta}, \boldsymbol{\Phi})$ Additionally predictions are generated by $y = g^{-1}(\boldsymbol{\Phi}\boldsymbol{\beta})$. Since $\boldsymbol{\Phi} \in \mathbb{R}^{n \times p}$ with $n < p$ there is a nullspace of $\boldsymbol{\Phi}$ with dimension at least $n - p$. Thus for any $u \in \text{null}(\boldsymbol{\Phi})$ we have $g^{-1}(\boldsymbol{\Phi}(\boldsymbol{\beta} + u)) = g^{-1}(\boldsymbol{\Phi}\boldsymbol{\beta}) = y$ and $f(\boldsymbol{\Phi}(\boldsymbol{\beta} + u), \boldsymbol{\Phi}) = f(\boldsymbol{\Phi}\boldsymbol{\beta}, \boldsymbol{\Phi}) = \mathcal{H}_{\boldsymbol{\beta}}$, which shows that the training predictions and the Hessian remain unchanged. $\square$

## G   TRAINING DETAILS

For the double descent experiments in Figures 1 and 2 we use neural network architectures from the following sources:

- CNNs from `https://gitlab.com/harvard-machine-learning/double-descent/-/blob/master/models/mcnn.py` but also include an option to vary the depth,
- ResNet18 from `https://gitlab.com/harvard-machine-learning/double-descent/-/blob/master/models/resnet18k.py`,

Specifically, we train with SGD with a learning rate of $10^{-2}$, momentum of 0.9, weight decay of $10^{-4}$ (thus corresponding approximately to a Gaussian prior of with variance 1000) for 200 epochs with a batch size of 128. The learning rate decays to $10^{-4}$ following the piecewise constant learning rate schedule in Izmailov et al. (2018) and Maddox et al. (2019), beginning to decay at epoch 100. We use random cropping and flipping for data augmentation — turning off augmentations to compute eigenvalues of the Hessian.

## H   DOUBLE DESCENT AND EFFECTIVE DIMENSIONALITY: FURTHER EXPERIMENTS

**Linear Models**   Although double descent is often associated with neural networks, we demonstrate similar behaviour with a linear model with a varying number of features: first drawing 200 data points $y \sim \mathcal{N}(0, 1)$ and then drawing 20 informative features $y + \epsilon$, where $\epsilon \sim \mathcal{N}(0, 1)$, before drawing $k - 20$ features that are also just random Gaussian noise, where $k$ is the total number of features in the model.[6] For the test set, we repeat the generative process. In Figure A.10 we show a pronounced double descent curve in the test error as we increase the number of features, which is mirrored by the effective dimensionality.

In Figure A.11, we plot the effective dimensionality against the test error for the linear model example in Section 5. A clear linear-looking trend is observed, which corresponds to the models that have nearly zero training error. The bend near the origin is explained by models that do not have enough

---

[6]From `https://github.com/ORIE4741/demos/blob/master/double-descent.ipynb`.

capacity to fit — therefore, their effective dimensionality is very small. We observe a similar trend for ResNets and CNNs.

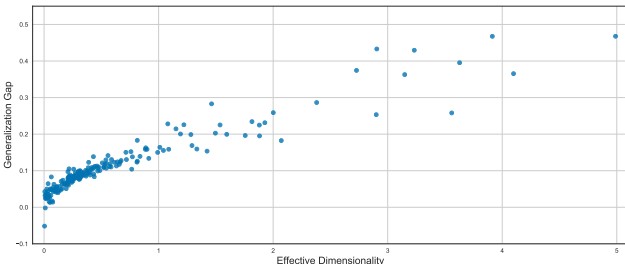

Figure A.11: Effective dimensionality plotted against generalization gap (test error − train error) for the linear model of Section 5. Note that all but the very smallest models with effective dimensionality track nearly linearly with generalization error.

**Small Neural Networks** Finally, we consider several further experiments on the two spirals problem to test the effects of increasing depth and width to serve as a sanity check for our results on both ResNets and CNNs. In Figure A.12, we fix the number of data points to be 3000, and vary the depth of the neural network (20 hidden units at each layer, ELU activations) using between one and 15 hidden units, training for 4000 steps as before. Here, we run each experiment with 25 repetitions and compute all of the eigenvalues of the Hessian at convergence (the largest model contains 6000 parameters). In the left panel, we see a pronounced double descent curve with respect to both effective dimensionality and test loss as we vary depth. In the right panel, we use the same data points, but use three hidden layer networks, varying the width of each layer between one and 30 units per layer. Here, we see only a monotonic decrease in both test error and effective dimensionality with increasing width not helping that much in terms of test error — the effective dimensionality is highest for the models with smallest size and slowly decreases as the width is increased. These results serve as a sanity check on our large-scale Lanczos results in the main text.

|  | Test loss | Test Error | Gen. Gap |
|---|---|---|---|
| $N_{eff}$(Hessian) | **0.9434** | 0.9188 | **0.9429** |
| PAC-Bayes | −0.8443 | −0.7372 | −0.8597 |
| Mag. PAC-Bayes | 0.7066 | 0.8270 | 0.6805 |
| Path-Norm | 0.5598 | 0.7216 | 0.5259 |
| Log Path-Norm | 0.9397 | **0.9846** | 0.9257 |

Table 1: Sample Pearson correlation with generalization on double descent for ResNet18s of varying width on CIFAR-100 with a training loss below 0.1.

|  | Test loss | Test Error | Gen. Gap |
|---|---|---|---|
| $N_{eff}$(Hessian) | 0.9305 | **0.9461** | 0.9060 |
| PAC-Bayes | −0.8619 | −0.7916 | −0.8873 |
| Mag. PAC-Bayes | 0.8724 | 0.9225 | 0.8330 |
| Path-Norm | 0.7996 | 0.7721 | 0.7511 |
| Log Path-Norm | **0.9781** | 0.9402 | **0.9602** |

Table 2: Sample Pearson correlation with generalization for CNNs of varying width and depth on CIFAR-100 with a training loss below 0.1.

**Deep Networks** In Figure A.15, we plot the effective dimensionality against the test error for the ResNet18s trained on CIFAR-100 with 20% data corruption. We show the sample Pearson correlation with respect to test loss, error, and the generalization gap for the same dataset of effective dimensionality along with PAC-Bayes, magnitude aware PAC-Bayes, path norm, and the logarithm of the path norm in Table 3. Finally, in Figure A.14, we plot the PAC-Bayes and magnitude aware

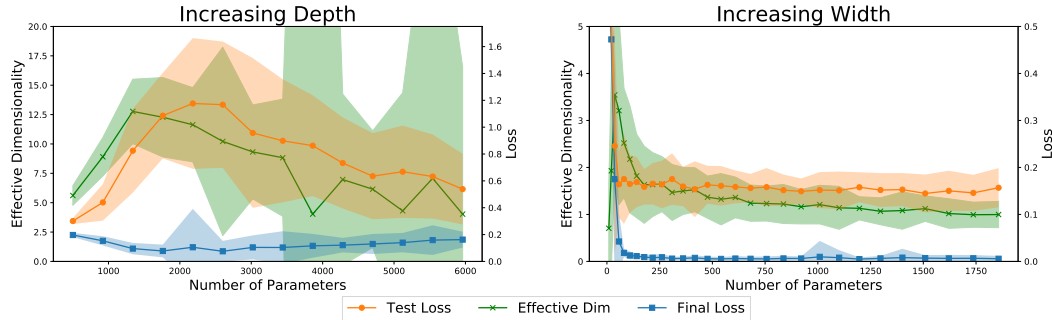

Figure A.12: **Left:** Increasing depth on the two spirals problem. Clearly seen is a double descent curve where the test loss first increases before decreasing as a function of depth. The effective dimensionality follows the same trend. **Right:** Increasing width on the two spirals problem. Here, increased width produces constant test performance after the training loss reaches zero, and the effective dimensionality stays mostly constant. Shading represents two standard deviations as calculated by 25 random generations of the spirals data.

|  | Test loss | Test Error | Gen. Gap |
|---|---|---|---|
| $N_{eff}$(Hessian) | 0.8772 | 0.87608 | 0.8578 |
| PAC-Bayes | 0.6644 | 0.6424 | 0.7758 |
| Mag. PAC-Bayes | 0.6933 | 0.6728 | 0.7956 |
| Path-Norm | 0.6722 | 0.6478 | 0.7937 |
| Log Path-Norm | **0.9585** | **0.9492** | **0.9836** |

Table 3: Sample Pearson correlation between generalization measures and the test loss, test error, and generalization gap for ResNet18s of varying width trained on CIFAR-100 with $20\%$ corruption that achieve a training loss below $0.1$. Among these models effective dimensionality and path-norm are most correlated with test loss, test error, and the generalization gap.

PAC-Bayes bounds on their raw scale for ResNet18s trained on CIFAR-100 and on the $20\%$ corrupted CIFAR-100 dataset. The trends are similar for the bounds across the two datasets, with the PAC-Bayes bound generally increasing as width increases. The magnitude aware PAC-Bayes bounds also act similarly, decreasing as width increases.

## I    EFFECT OF NUMBER OF EIGENVALUES USED

[Figure A.16 shows the model effective dimensionality for small MLPs trained on the two spirals problem from Section A.2 over $10$ random initializations when we vary how many eigenvalues are used in the computation. From this experiment we see first that provided a large enough amount of the dominant eigenvalues are used, effective dimensionality is insensitive to exactly how many eigenvalues are considered. Secondly, we observe that upon repeatedly initializing and training networks the effective dimensionality generally falls in the same range of values, up to some randomness in that we may find classifiers of different performance in each training run.]

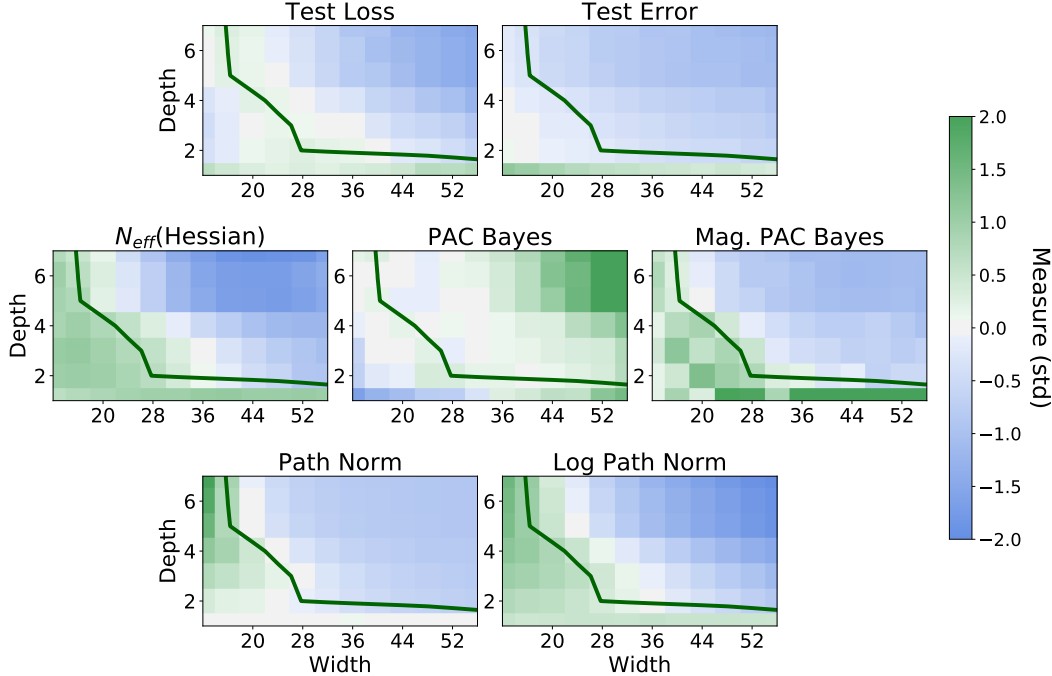

Figure A.13: Plots of all generalization errors considered by width and depth. The green curve separates models that achieve below 0.1 training error. Additionally, we standardize all measures and test error to 0 mean and unit variance for comparison.

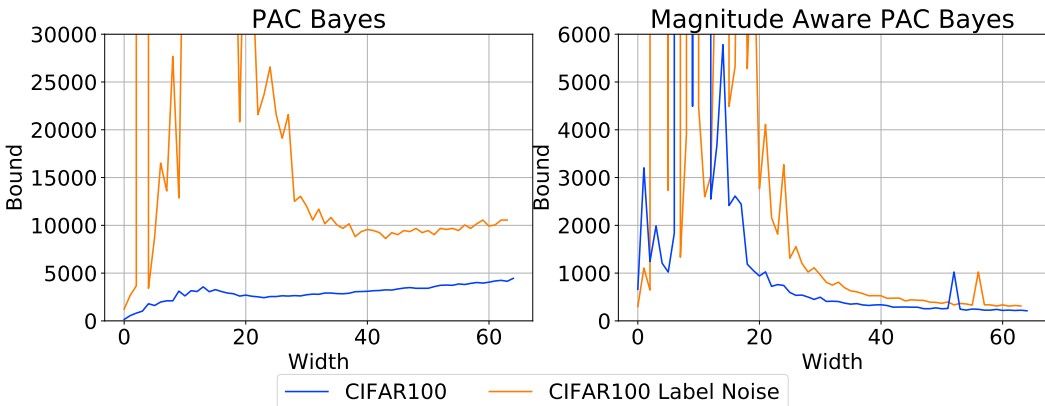

Figure A.14: PAC-Bayes and magnitude aware PAC-Bayes bounds displayed on their raw scale for both CIFAR-100 and the 20% corrupted CIFAR-100. All have a significant spike near a width of 20 which is when training loss first reaches zero. The standard PAC-Bayes bounds increase as width increases, while the magnitude aware ones seem to decrease towards zero.

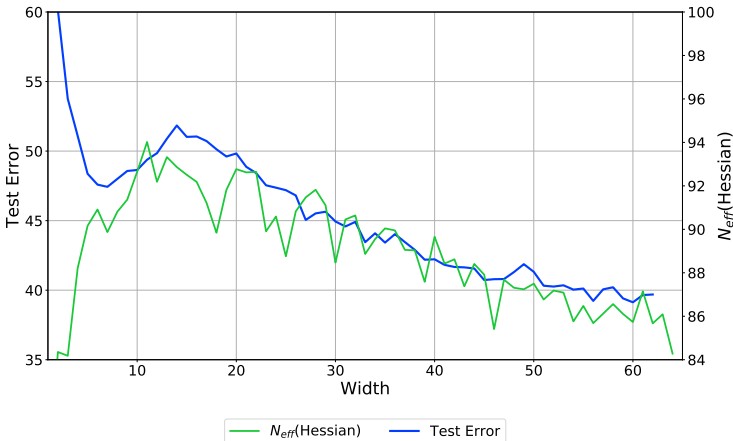

Figure A.15: Double descent with respect to test error as demonstrated on ResNet18 on CIFAR-100 with 20% corruption. The effective dimensionality again tracks the double descent curve, this time present in the test *error* rather than test *loss*.

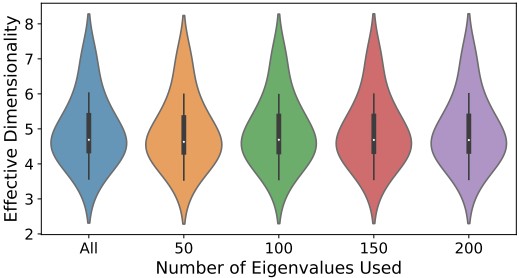

Figure A.16: **[Violin plots of effective dimensionality over 10 random initializations of the two spirals problem when the top $k$ eigenvalues are used to compute the model effective dimensionality].**

