# OpenReview forum: "Rethinking Parameter Counting: Effective Dimensionality Revisited"
_ICLR.cc/2021/Conference — Reject_

### Official Review · AnonReviewer1 · 2020-10-19
**A borderline paper with interesting insights?**

**Rating:** 6
**Confidence:** 3

**Review:**

**Summary**: In this article, the authors revisited the idea of *effective dimensionality* as a complexity measure for large-scale machine learning systems, and in particular, modern deep neural networks. Theoretical arguments were provided for linear and generalized linear models (Theorem 4.1 and 4.2). Connections were made between the proposed effective dimensionality and the double descent phenomenon, width-depth trade-off, function-space homogeneity, and other generalization measures in the literature. Experiments on linear models as well as deep networks (ResNet18) were provided to support the effectiveness of the proposed metric.

**Strong points**: The authors revisited the idea of *effective dimensionality* as a complexity measure for large-scale machine learning systems, and in particular, modern deep neural networks. Theoretical arguments were provided for linear and generalized linear models. Insightful discussions were made on the connection between the proposed effective dimensionality and the double descent phenomenon, width-depth trade-off, function-space homogeneity, and other norm- or flatness-based generalization measures. The paper is in general well-written.

**Weak points**: The presentation of the article can be significantly improved. The contribution, from either a theoretical (Theorem 4.1 and 4.2 on Bayesian linear models with Gaussian prior, with generalized linear models in the appendix) or an empirical (ResNet18 on CIFAR-100) perspective, seems not enough for a clear accept.

**Recommendation**: On account of the theoretical or empirical contributions of this work, I find this paper somewhat borderline. Nonetheless, according to the strong points I mentioned above and in particular, the interesting and novel insights offered by this paper into the understanding of deep neural nets, I'm more leaning toward an acceptance.

**Detailed comments**:

* P3 Section 2 "matrix of second derivatives of the loss, $H_{\theta} = - \nabla \nabla_{\theta} \mathcal L(\theta, \mathcal D)$": what does $\mathcal D$ mean here?
* P3 Section 2.1 "This increase in curvature of the loss that accompanies certainty about the parameters leads to an increase in the eigenvalues of the Hessian of the Growth in eigenvalues of the Hessian of the loss corresponds to increased certainty about parameters": is this a **general** claim, how is this theoretically/empirically supported? And it is not clear, at least to me, how the same intuition built here extends to general cases as claimed by the authors below Figure 4 in P4.
* are the (Hessian) eigenvalues assumed to be all **positive** in the definition of effective dimensionality? This may not be the case for neural networks.
* "Therefore, effective dimensionality explains the number of parameters that have been determined by the data": at this point (of the article), it is not yet clear to me how the effective dimensionality defined above is connected to the data.
* P4 Practical Computations: there exists a library called "PyHessian: Neural Networks Through the Lens of the Hessian" that can perform many eigenspectrum-based computations of the Hessian of deep neural nets, which might help to conduct experiments beyond the first $100$ eigenpairs of the Hessian, though honestly, I have not tried it myself.
* Theorem 4.2 states the "function-space homogeneity" of a subspace of the Hessian, in the sense of the training prediction, how does this affect the test performance of the model?
* Section 5.1: "tracks remarkably well with generalization — displaying the double descent curve that is seen in the test loss": this is not entirely true, the first (local) minimum of the effective dimensionality and of the test loss appear at a relatively different width. It seems to me that the proposed metric is more "accurate" for (nearly) interpolation models (i.e., models with zero or low training loss): this is also seen at the bottom of the left plot of Figure 2 where the effective dimensionality (with high training loss) is low, while it is not the case for test loss.

---

> ### Author Response · Authors · 2020-11-19
> **Response to Reviewer 1**
>
> We appreciate your thoughtful and supportive review.  We hope that our response and revisions have helped alleviate your concerns and have helped to connect the theoretical and empirical contributions. We want to emphasize that our contributions are not purely theoretical, or empirical, but both --- and that both contribute synergistically to the paper.
>
> We also want to emphasize that the paper is making several very deep and timely contributions that are all interconnected: (1) for the first time tracking double descent with a generalization metric, (2) exploring generalization as a function of depth (which has largely been ignored, with width instead as a focus, despite the practical significance of depth for generalization), (3) providing insights into effective dimension as model compression and links to Bayesian posterior contractions, (4) providing important contributions to the pervasive parameter counting narrative in contemporary deep learning; (5) explaining why subspace compression methods in deep learning are effective through the lens of effective dimensionality (these methods have been highly mysterious despite their practical success) by exploring properties of function-space; (6) showing that effective dimension actually provides a very competitive generalization measure relative to several generalization measures that have been isolated as high performing in recent literature. We hope you can consider the importance, timeliness, and synergy of these contributions, in considering your final assessment.

---

> > ### Author Response · Authors · 2020-11-19
> > **Response to Reviewer 1 (cont.)**
> >
> > Thank you for the detailed comments, below are our responses to them, which we’ve clarified in the updated version.
> >
> > Q: P3 Section 2, what does \mathcal{D} mean here?
> > A: It means the training dataset.
> >
> > Q: "Therefore, effective dimensionality explains the number of parameters that have been determined by the data: at this point (of the article), it is not yet clear to me how the effective dimensionality defined above is connected to the data.”
> > A: The Hessian of the loss is dependent on the data through the loss function. Thus the eigenvalues of the Hessian of the loss implicity depend on the data. We’ve clarified this point in the updated version.
> >
> > Q: are the (Hessian) eigenvalues assumed to be all positive in the definition of effective dimensionality?
> > A: At a local minimum the eigenvalues will all be non-negative. For deep neural networks the Hessian for stationary points of SGD can in principle have negative eigenvalues, but experimentally we find that they are extremely small in magnitude or non-existent when we are considering the Hessian of the log likelihood plus the Gaussian prior on the parameters (which increases the eigenvalues). Moreover, in practice, only the 50 or so largest eigenvalues actually contribute to the computation of the effective dimension, even when there are more than 10 million eigenvalues. So in practice it is safe to assume that the eigenvalues are positive or small enough in magnitude as to not affect the effective dimensionality.
> >
> > We have included a small experiment showing that upon training 10 models on the two spirals dataset the effective dimensionality is unchanged regardless of how many of the eigenvalues we consider (including using all eigenvalues). Furthermore in all models considered approximately 95% of the more than 5000 eigenvalues are greater than -0.01.
> >
> >
> >
> >
> > Q: Theorem 4.2: How does function space homogeneity affect test performance of the model?
> >
> > A: The function space homogeneity suggests that despite the model being “overparametrized” the functional solutions it provides can be represented in a relatively low dimensional subspace --- meaning the model creates a compression of the data that is likely amenable to good test set generalization. It is a compelling example of how one must be careful in treating parameter counting as a proxy for understanding generalization properties.
> >
> > Incidentally, the difference in function-space homogeneity between test and train sets is not that large provided that the test and train sets come from the same distribution. We point out that the empirical perturbation analysis as shown in Figure 6 is on a test set. We train only on the white and black data points as shown in the left panel, while testing on a dense grid of [0,1]^2 data as shown in the center and right panels. Even on the test set perturbations in the null space of the Hessian do not significantly affect predictions. We have also updated the appendix to include Figure A.9 which shows that the functions are similarly homogeneous on the CIFAR10 test set as well.
> >
> > Q: P3 Section 2.1: Why should the increased certainty about the parameters yield an increase in the eigenvalues of the Hessian and vice versa?
> > A: In the example in Figure 3, the problem is one dimensional so the Hessian corresponds to the inverse of the variance of the posterior distribution on the mean parameter. As we observe more data, the variance of the posterior shrinks and so the inverse variance and the Hessian eigenvalue increases.
> >
> > Similarly, for Bayesian linear models, the Hessian corresponds to the posterior covariance matrix. As we observe more data, the posterior covariance matrix concentrates around a single point, and the Hessian eigenvalues grow.
> >
> > We point to references (Sagun et al, and Ghorbani et al) that find that the eigenvalues of the Hessian tend to grow throughout training the model, indicating that the model is becoming more certain about its predictions (as well as the parameters themselves).
> >
> > Q: PyHessian.
> > A: We thank the reviewer for the pointer to PyHessian, which we will include a reference to on updating the paper. PyHessian, like our approach, as well as the approach of Ghorbani et al. (code: https://github.com/google/spectral-density), all use Lanczos and implicit Hessian vector products to compute a pre-specified number of eigenvalues of the Hessian.
> >
> > Q: Section 5.1 Effective dimensionality doesn’t track well if the training loss is high.
> > A: Indeed, as we explain above, effective dimension works best for model comparison when the models being compared both have similarly low training loss. Then the models can be viewed as essentially lossless compressions of the data -- and the one that provides the best compression, and hence has the lowest effective dimension, will tend to provide the best generalization.
> > We’ve updated the paper to be a bit more careful about these claims to state “tracks remarkably well with generalization amongst models with low training loss”.

---

### Official Review · AnonReviewer3 · 2020-10-28
**Interesting insights into generalization in deep learning based on the effective dimensionality**

**Rating:** 6
**Confidence:** 3

**Review:**

# Summary

The paper applies the effective dimensionality (introduced by MacKay, Gull and others) to study the generalization properties of large probabilistic models. Effective dimensionality is the number of parameters determined by the data (derived from the curvature of the posterior at the MAP estimate), and shown to be more informative than simple parameter counting. After demonstrating the usefulness of the effective dimensionality, the authors study double descent observed when training deep nets of increasing width/depth. The authors argue that double descent is an artifact that can be understood by studying the effective dimensionality of the model. They take a detailed look at width-depth trade-offs using numerical experiments. Moreover, they compare the effective dimensionality with other generalization measures and find a superior performance.

# Assessment

Overall, I enjoyed reading the paper. Most of the time it is well-written and provides some new insights into questions concerning generalization in probabilistic models. So I'm tending towards acceptance, but there are several problems with the current version of the paper.

## Pros

- Effective dimensionality is shown to be a useful quantity to understand generalization properties of probabilistic models in particular deep neural nets.

- Effective dimensionality helps us understand width-depth trade-offs in deep learning.

- Effective dimensionality is a metric for generalization solely based on the training data.

## Cons

- Effective dimensionality rests on the Laplace approximation (of the log posterior) which fails for multimodal posteriors.

- Organization of material is suboptimal. For example, section 5 refers to figures 1 and 2, which already has been discussed in the Introduction. Occasional sloppiness in notation and wording.

- Theorems are rather elementary; I'm not sure whether they should be highlighted as theorems. Validity for neural nets is only hypothesized and demonstrated empirically. Any analytical results?

# Comments / Questions

- The organization of paper a bit difficult to follow. Central results (Figs. 1 and 2) are already shown early in the paper and later explained in more detail...

- There is no explanation as to why the effective dimensionality decreases with increasing width/depth (as shown in Figs. 1 and 2). Why do we see a decrease in effective dimensionality?

- How stable is the calculation of effective dimensionality across different training runs?

- If you are only computing the 100 largest eigenvalues, the effective dimensionality will always be smaller or equal to 100. Why do you restrict the effective dimensionality to a maximum of 100 in most of your experiments?

- How much sense does effective dimensionality make for multimodal posteriors?

- The use of the notion "effective dimensionality" is sometimes confusing: On the one hand, it is a property of any positive semi-definite matrix (Eq. 2). On the other hand, most of the time "effective dimensionality" is implicitly understood as "the effective dimensionality of the model" (i.e. $N_{eff}$ of the Hessian of minus log posterior). I would prefer to use $N_{eff}$ when you talk about the effective dimensionality of a specific matrix and "effective dimensionality of the model" when you mean $N_{eff}$ of the Hessian of minus log posterior. An instance where your ambiguous use of "effective dimensionality" leads to confusion can be found on page 5: "For Bayesian linear models, the effective dimensionality of the parameter covariance is the inverse of the Hessian" -- it's not clear to me what you mean by that...

- Page 2: "we expect models with lower effective dimensionality to generalize better" -- why?

- Page 3: Right before eq. (1): You specify "$y \sim \mathcal N(f=\Phi^T\beta, \sigma^2)$". What do you mean by "$f=...$"? Shouldn't "$f=$" be removed? Moreover, in the theorems and their proofs you work with the transpose of $\Phi$ since the model is $y\sim \mathcal N(\Phi\beta, \sigma^2 I_n)$... Also the prior "$\beta \sim \mathcal N(0, \alpha^2 I_N)$" is not consistent with the theorems and the appendix where $k$ (sometimes $p$) is used to indicate the number of parameters.

* Page 3: Right after eq. (1): "where $\lambda_i$ are the eigenvalues of $\Phi\Phi^T$, the Hessian of the log likelihood". Strictly speaking $\Phi\Phi^T/\sigma^2$ is the Hessian of minus log likelihood. Once again, please use one consistent model and notation in the main text and the appendix (either $y\sim \mathcal N(\Phi^T\beta, \sigma^2 I_n)$ or $y\sim \mathcal N(\Phi\beta, \sigma^2 I_n)$ -- I prefer the latter in which case the Hessian of minus log likelihood is $\Phi^T\Phi/\sigma^2$ rather than $\Phi\Phi^T/\sigma^2$...) and carefully adapt all expressions that are affected by your choice.

- Page 6, Equation 3: Your Occam factor scales with $1/\sqrt{\mathrm{det}(\mathcal H_\theta)}$. What happens if the Hessian is singular (which is bound to happen for overparameterized models)? Comparison with MacKay's definition reveals that you should replace $\mathcal H_\theta$ with $\mathcal H_\theta + I_k / \alpha^2$, which resolves the trouble with singular Hessians of minus log likelihood.

- Also in other instances, you tend to focus on the Hessian of minus log likelihood, whereas MacKay looks at $A = - \nabla\nabla \log p(\theta|\mathcal D, \mathcal M) = \mathcal H_\theta + I_k/\alpha^2$ (Hessian of minus log posterior / posterior curvature / inverse posterior covariance). I find this confusing. I think it would help to always clearly state, if you talk about the Hessian of minus log likelihood or the Hessian of minus log posterior.

- Appendix: "In the overparameterized regime, $k > n$, with linearly independent features we have that has rank at most $k$" (page 18). This is incorrect, the rank is at most $n$. In both proofs: Why not argue using the SVD of the feature matrix? If $\Phi = U\Lambda V^T$ with column-orthogonal matrices $U, V$ such that $U^TU=V^TV = I_r$ (where rank $r \le \min(n,k)$), we have $\Phi\beta = U\Lambda V^T\beta$. Use projectors $P=VV^T$ and $P_\perp=I_k - VV^T$ to decompose $\beta$ into contributions that affect the prediction, $P\beta$, and perturbations that do not change the prediction, $P_\perp\beta$ (also: $\|\beta\|^2 = \|P\beta\|^2 + \|P_\perp\beta\|^2$). $P$ projects into an $r$-dimensional linear subspace, $P_\perp$ projects into the orthogonal space (dimension: $\max(n,k) - r$) of neutral perturbations: $\Phi\beta=\Phi(P + P_\perp)\beta = \Phi P\beta$ since $\Phi P_\perp = 0$.

# Minor

* Page 2: symbol $\mathcal L$ (log likelihood) is not or only implicitly defined in the text

* Page 5, Fig. 5, left panel: The tick labels on the right axis ($N_{eff}$) are very small (ranging from 0 to 4). Is this correct?

* Page 6, Equation 3: It would be helpful to explain all symbols (i.e. $p(\theta_{MP}|\mathcal M)$ is the prior evaluated at the MAP estimate...)

---

> ### Author Response · Authors · 2020-11-19
> **Response to Reviewer 3**
>
> Thank you for your detailed, thoughtful, and supportive review! You have brought to light a number of helpful ways we can bring more clarity to the paper -- which we have incorporated in an updated version.
>
> We want to emphasize that we believe our exploration of effective dimensionality makes a very timely and significant contribution, (1) for the first time tracking double descent with a generalization metric, (2) exploring generalization as a function of depth (which has largely been ignored, with width instead as a focus, despite the practical significance of depth for generalization), (3) providing insights into effective dimension as model compression and links to Bayesian posterior contractions, (4) providing important contributions to the pervasive parameter counting narrative in contemporary deep learning; (5) explaining why subspace compression methods in deep learning are effective through the lens of effective dimensionality (these methods have been highly mysterious despite their practical success) by exploring properties of function-space; (6) showing that effective dimension actually provides a very competitive generalization measure relative to several generalization measures that have been isolated as high performing in recent literature. We hope you can consider the importance, timeliness, and synergy of these contributions, in considering your final assessment.
>
> We appreciate that you (and other reviewers) noted that the paper is generally well-written. While we agree several stylistic decisions can have both pros and cons, we do note that the decision to have figures 1 and 2 early-on in the paper was carefully considered, rather than arbitrary. The rationale was to have some of the key results appear early, so that a reader could become quickly engaged with the paper, and have a clear sense of what it’s about --- it sets up much of the material that follows. At the same time, it’s hard to have all the details in an introduction, and so we provided further detail later in the paper --- including additional comparisons related to these results. To improve clarity, we now signpost additional related material that comes later in the text.

---

> > ### Author Response · Authors · 2020-11-19
> > **Response to Reviewer 3 (cont.)**
> >
> > We appreciate your description of strengths. We would like to open with responses to the cons you have listed.
> >
> > - We agree that the effective dimensionality [ED] looks at local behaviour within a mode. However, we would appreciate if you could consider three key points: (1) as far as we can tell, no generalization measure used in modern deep learning accounts for multimodal posteriors, so it may be unfair to hold this against ED; (2) ED does not “fail” for multimodal posteriors… indeed, all of the neural network posteriors in our experiments would be multimodal, but ED still does a relatively good job of model comparison. This is the case for the same reason it is possible to train a neural network with SGD, despite multimodality, and reliably find reasonable generalization. While the posterior is multimodal, most of the modes easily discoverable by SGD provide a similar level of performance, albeit sometimes complimentary solutions; (3) a unimodal measure is applicable for model comparison to an overwhelming majority of models in practice, which are typically trained with optimization (which converges to a single mode even if the loss surface is multimodal), or unimodal marginalization. A multimodal measure would mostly only be applicable if we are comparing between models that are a result of multi-basin marginalization. Our intention is to show that effective dimension can be informative for comparing models which have been trained in a standard way, which is typically optimization or unimodal marginalization.
> >
> > - While we agree the theorems are relatively straightforward, we do not believe that should be held against the paper. There are many contributions in the paper, and in fact we do not reference the theorems as core contributions of the paper in the introduction. Moreover, the theorems do combine synergistically with the content: we show that many of the results that can be proven for linear models or generalized linear models hold for neural networks. We primarily use these theorems as stepping stones towards gaining insights into the behavior of large neural networks. We would also posit in this context that being “straightforward” is arguably an advantage, and that relevance and impact are more important than complexity in results.
> >
> > - We appreciate that you (and other reviewers) noted that the paper is generally well-written. While we agree several stylistic decisions can have both pros and cons, we do note that the decision to have figures 1 and 2 early-on in the paper was carefully considered, rather than arbitrary. The rationale was to have some of the key results appear early, so that a reader could become quickly engaged with the paper, and have a clear sense of what it’s about --- it sets up much of the material that follows. At the same time, it’s hard to have all the details in an introduction, and so we provided further detail later in the paper --- including additional comparisons related to these results.

---

> > > ### Author Response · Authors · 2020-11-19
> > > **Response to Reviewer 3 (cont.)**
> > >
> > > In response to the comments and questions:
> > >
> > > - Effective dimensionality decreasing with model size:
> > > We do actually have an explanation in section 5.1: “as the dimensionality of the parameter space continues to increase past the point where the corresponding models achieve zero training error, flat regions of the loss occupy a greatly increasing volume, and are thus more easily discoverable by optimization procedures such as SGD. These solutions have lower effective dimensionality, and thus provide better compressions of the data, as in Section 4.3, and therefore better generalization”.  Moreover, larger models (models which are wider and deeper) tend to have larger capacity and be more expressive, and thus can find higher performing compressions of the data (e.g. they contain good subnetworks), which leads to decreased effective dimensionality. We will clarify these points earlier on.
> > >
> > > - Stability to training runs + computing dominant eigenvalues: Effective dimensionality is very stable across different training runs. The ranking of models by effective dimensionality stays consistent with generalization in repeated trials. In practice for large neural networks we see an eigenspectrum of the Hessian that contains a small number of large eigenvalues followed by many eigenvalues that are approximately 0. To capture the general behavior of the eigenspectrum of the Hessian we need only compute this small number of eigenvalues. Therefore to save on computations while retaining an accurate estimate of effective dimensionality we compute the top 100 eigenvalues of the Hessian.
> > > Inspired by your question, we have also added Figure A.16 Appendix as an example of stability of training runs and the consistency of effective dimension when using more than 50 eigenvalues.
> > >
> > > - For cases where we only examine a single mode in the loss surface (i.e. ML, MAP, Laplace approximations, [1], [2]), effective dimensionality is consistent with the construction of the model. Extending concepts like effective dimensionality to models that consider multiple modes in the loss surface is an interesting direction for future work, but quite different from the intention of our paper, which is in part to show how effective dimension can be an informative metric for models trained in a standard way. We have further comments about ED and multimodal posteriors in the opening of our response.
> > >
> > > - Clarity of effective dimensionality of a matrix vs of the model: Thank you for the question. We have clarified this point in the paper, keeping consistent distinctions between effective dimensionality as a function of a matrix, and effective dimensionality of the Hessian of the loss.

---

> > > > ### Author Response · Authors · 2020-11-19
> > > > **Response to Reviewer 3 (cont.)**
> > > >
> > > > - Effective dimensionality and generalization on page 2:
> > > > When two models have the same training loss they can be viewed as providing a compression of the training data at the same fidelity, in which case the model which has the lower effective dimensionality, and thus provides the better compression  --- capturing more regularities --- will tend to generalize better. Flatter solutions lead to lower effective dimensionality and also have been connected in many studies with better generalization [1, 2, 3, 4]. We also show in Section 4.3 that lower effective dimensionality provides a better Occam’s factor and shorter minimum description length.
> > > >
> > > > - If we do not hold training loss constant when comparing models, then it is possible a model could achieve a lower effective dimension simply by extracting less information in the training data, which would typically not be predictive of generalization.
> > > >
> > > > - We will make more clear the caveat that it is most interpretable to compare effective dimensionality for models with similar training loss. We also now signpost in the introduction our explanations for why lower effective dimensionality can lead to better generalization in section 4.2 and 4.3, with respect to minimum description length and Occam factors, and connections between flatness and generalization.
> > > >
> > > > - Regarding the notation of $\Phi^T \beta$: You are correct --- we have made the necessary changes to keep notation consistent.
> > > >
> > > > - Regarding page 6, equation 3: $\mathcal{H}_{\theta}$ is the Hessian of the log posterior, not the likelihood, so we should not have issues with a zero determinant. We have made sure to clarify this in the text.
> > > >
> > > >
> > > > - Hessian of the negative log likelihood: We only mention the effective dimensionality of the Hessian of the likelihood in the context of Equation 1. Typically we are looking at the Hessian of the loss which we note in Section 2.1 is the negative log posterior. We have clarified this point in the text to stave off further confusion.
> > > >
> > > >
> > > > - Appendix page 18: Good catch --- this was a typo we have now fixed in the paper.
> > > >
> > > > [1] Averaging weights leads to wider optima and better generalization, Izmailov et al. 2018
> > > > [2] A simple baseline for bayesian uncertainty in deep learning, Maddox et al. 2019
> > > > [3] On Large-Batch Training for Deep Learning: Generalization Gap and Sharp Minima, Keskar et al. 2017
> > > > [4] Flat Minima, Hochreiter and Schmidhuber 1997

---

### Official Review · AnonReviewer2 · 2020-10-29
**Interesting avenue, but requires improvements**

**Rating:** 4
**Confidence:** 4

**Review:**

This paper explores the effective dimensionality of the Hessian as a possible explanation for why neural networks generalize well despite being massively overparametrized.

While I concur with the intuition, I think in the current state of the paper, some points could be improved and clarified.

### Relationship between Hessian and posterior covariance

While you mainly reason in the Bayesian framework about the posterior, it seems that networks in fig 1, 6, 7 are trained using ML. So why would the Hessian of the ML estimator relate to the covariance of the posterior?

### Theorem 4.1

Theorem 4.1 shows that even with $k \gg n$ parameters, there are only $n$ directions in which the posterior covariance changes from the prior. But the rest of the discussion shows that the effective dimension actually decreases, which is not captured by your theorem. In this regard, I consider this theorem as an illustration of why the effective dimension does not increase with increasing number of parameters, a statement that is weaker than saying that the effective dimension actually decreases.

Therefore, we can say that your argument for advocating in favor of using the effective dimension as a proxy for generalization is mainly empirical. Then I would have appreciated a more thorough ablation study, that would demonstrate that the correlation is still occuring while varying other hyperparameters.

### Figures 4

Fig 4: can you precisely state what is plotted, i.e. for a fixed $z$ why do we have several datapoints?

### Conclusion

As already said in the beginning I really like the idea of effective dimension playing an important role in generalization. I however think that relationship between the hessian of ML estimator and the covariance of the posterior, as well as the empirical study, should be improved before this is published.

---

> ### Author Response · Authors · 2020-11-19
> **Response to Reviewer 2**
>
> Thank you for the thoughtful remarks. Thank you also for the specific questions, which we believe are we able to directly address below. We would value it if you would consider updating your assessment in light of our response.
>
> Relationship between Hessian and posterior covariance: We offer a view of the Hessian of the loss that is compatible with *both* Bayesian and maximum likelihood frameworks. In both cases we are considering a posterior which is proportional to a likelihood times a prior. In both cases the effective dimensionality describes the number of parameters determined by the data in terms of the number of sharp directions in the posterior. In the maximum likelihood framework the negative log posterior acts as the loss surface for optimization, and the prior acts as a regularizer. In the Bayesian case the parameter determination by the data can be understood as contraction of the posterior from the prior, which we show is consistent with our interpretation of effective dimensionality.
>
> Moreover, in the case of Bayesian linear regression with a Gaussian prior, the Hessian of the maximum likelihood estimator \hat \beta is exactly the inverse posterior covariance matrix over the parameters, a fact we explicitly use in the proof of Theorem 2 in the Appendix (see Appendix F.2). This algebraic relationship has the very nice property that it makes the Laplace approximation to the marginal likelihood (Eq. 3) exact. Furthermore, we demonstrate empirically in Figure 5 (right) that the effective dimensionality of the posterior covariance of small Bayesian NNs acts in an inverse fashion (it decreases as the number of data points increases) to the effective dimensionality of the Hessian matrix (which increases as the number of data points increases). The empirical result suggests that for Bayesian NNs the Hessian of the posterior at the ML estimator is very closely related to the inverse posterior covariance matrix.
>
> Theorem 4.1: We agree that Theorem 4.1 in the paper does not imply that effective dimensionality should decrease as the number of model parameters grows. But we do see the theorem has harmonizing with our empirical results and the general narrative of the paper. This theorem serves to highlight an immediate failure of the approach of parameter counting: in the overparameterized setting there will be many directions in which the parameters have not been determined by the data. We are using the theorem as a stepping stone to show that our intuitions from linear models (i.e. undetermined parameter directions) should hold for neural networks --- which we do see happens in practice, with our experiments.
>
> We would like to emphatically clarify, however, that the argument for using effective dimension as a proxy for generalization is not only empirical. Theorem 4.1 is one of many results in the paper. The reasoning for why effective dimension should be a good proxy for generalization is also given in section 4.3, where we show that lower effective dimension leads to better Occam factors and connects with a lower minimum description length. There are also many results in the literature connecting flatness with generalization --- arguing that flatter solutions, which by definition will have lower effective dimension, often correspond to better generalization. We will make this point more clear in the text [1,2,3,4]. But we have also added an ablation study, inspired by your comments in which we train a number of networks of varying widths on CIFAR10 with different learning rates and weight decays and compare effective dimensionality and test accuracy.
>
> Figure 4: This figure shows the effective dimensionality of the collection networks with near zero training loss from Figure 2 for a range of regularization parameters z. In the revised version of the paper, we’ve updated the figure caption to better explain the figure. This is an important result, which we included to be especially thorough empirically, showing that the qualitative comparison between models given by the effective dimension is fairly robust to settings of ‘z’.
>
> [1] Averaging weights leads to wider optima and better generalization, Izmailov et al. 2018
> [2] A simple baseline for bayesian uncertainty in deep learning, Maddox et al. 2019
> [3] On Large-Batch Training for Deep Learning: Generalization Gap and Sharp Minima, Keskar et al. 2017
> [4] Flat Minima, Hochreiter and Schmidhuber 1997

---

### Official Review · AnonReviewer4 · 2020-10-29
**Official Blind Review#4**

**Rating:** 5
**Confidence:** 3

**Review:**

summary:
This paper provide a unified view of the generalization ability in the Bayesian deep learning framework through the effective dimensionality. The authors claim that some phenomenon in the deep learning such as generalization in #parameter >> #data settings, and double descent can be explained by the effective dimensionality.

Although the theorem and experiments in the paper suggest that some of these properties can be explained by effective dimensionality, it is insufficient to convince that it substitutes other measures such as parameter counting.
For example, in Figure 2 abd 7, the effective dimensionality shows a very different behavior from test loss and test error when the width is small.　In other words, effective dimensionality does not seem to account for the first descent in Figure 2 and Figure 7 (although it follows the second descent well).

typos

- p.17 in MEASURING POSTERIOR CONTRACTION IN BAYESIAN GENERALIZED LINEAR MODELS

	-- The numerator in the second line of equation (11): 1 - \alpha^2(\lambda_i + \alpha^-2) -> \alpha^2(\lambda_i + \alpha^-2) - 1?

- p.18 in F.1 PROOF AND EXTENSIONS TO THEOREM 4.1

  -- "...the posterior distribution of \beta has an p-k directional subspace..." -> "...the posterior distribution of \beta has an k-n directional subspace..."?

  -- "Therefore, the posterior covariance has p-n directions..." -> "Therefore, the posterior covariance has k-n directions..."?

---

> ### Author Response · Authors · 2020-11-19
> **Response to Reviewer 4**
>
> Thank you for your review. There appears to be a key misunderstanding in your concern which we would like to clarify.
>
> We emphasize that effective dimension can only be used for model comparison ***when we are comparing models of similar training loss***. Each model can then be viewed as providing a compression of the training data at the same fidelity, in which case the model that has the lower effective dimensionality (ED), and thus provides the better compression, will tend to generalize better. If we do not hold training loss constant, then it is possible a model could achieve a lower effective dimension simply by extracting less information in the training data, which would typically not be predictive of generalization. We can also see this perspective in our discussion of connections with the marginal likelihood in Section 4.3, where there is both a model fit and complexity penalty term. A model with low ED but poor model fit (e.g. training loss) will have a poor marginal likelihood. Two models with similar model fit but one with lower ED will have a better marginal likelihood, typically leading to better generalization.
>
> We highlight this caveat about controlling for training loss for comparison in several places in the text. E.g., “we see that once a model has achieved low training loss, the effective dimensionality, computed from training data alone, replicates double descent behaviour” (page 2); “for models with low training loss (above the green partition), the effective dimensionality closely tracks generalization performance for each combination of width and depth” (page 2); “The green curve separates models with near-zero training loss” (page 2); “As the dimensionality of the parameter space continues to increase past the point where the corresponding models achieve zero training error,flat regions of the loss occupy a greatly increasing volume” (page 7); “In the region of near-zero training loss, separated by the green curve, we see effective dimensionality closely matches generalization performance.” (page 7).
>
> We have updated the text to more explicitly explain why effective dimensionality should generally be used to compare models with similar training loss, as in the second paragraph of our response here. If this condition is met, effective dimension certainly does provide a compelling alternative to parameter counting, and several state-of-the-art generalization metrics, which we can see in the results of Figures 1, 2, 5, and 7.
>
> We believe this response fully addresses your concern, and would therefore appreciate it if you would consider substantially raising your score. Feel free to let us know if you have any further questions.

---

### Author Response · Authors · 2020-11-19
**Paper updates**

We thank the reviewers for helpful and encouraging comments! We have made the following updates to the paper, and also provide detailed responses to the individual reviewers in separate posts.

Clarifications
- We have clarified that the effective dimension should be used for model comparison when the models being compared both have similarly low training loss. Then the models can be viewed as low-loss compressions of the data --- and the model that provides the best compression, and hence has the lowest effective dimension, will tend to provide the best generalization. If there is high loss, or models with different training loss, then the level of compression is less relevant, because the models have not necessarily learned very much from the training data. This explains why effective dimension closely tracks generalization for both double descent and width depth tradeoffs in the region of low training loss (and much better than parameter counting, which suggests the opposite trend).
- We’ve updated the caption to Figure 4 to make it clear that each point is a model of varying width and depth from Figure 2, while the color on each point is a different parameter, z, for computing the effective dimensionality.
- We’ve made the distinction between effective dimensionality as a function of matrices and the model effective dimensionality clearer, and signposted that when we are talking about generalization performance we are referring to effective dimensionality of the Hessian of a trained model.
- We’ve fixed usage of the shape of the features matrix \Phi throughout.
Emphasized that the Hessian is dependent on the dataset the model is trained on in Section 2.
We’ve cleaned up the typos in the proof of Theorem F.1 in the Appendix.

New experiments:
- We’ve added Appendix Figure A.9, which shows function space homogeneity in directions of the Hessian which have small eigenvalues on the test set of CIFAR10 for a CNN model.
- We’ve also added Figure A.16, which demonstrates on a MLP that the number of eigenvalues used to compute the effective dimensionality of the Hessian is quite robust to the number of eigenvalues used.


Additional updates:
- We’ve appended the Appendix to the main pdf file.
- We’ve included discussion of PyHessian and of the existence of negative eigenvalues in Section 2.

---

### Decision · Program_Chairs · 2021-01-07
**Final Decision**

**Decision:**

Reject

**Comment:**

The authors re-state Mackay's definition of effective dimensionality and describe its connections to posterior contraction in Bayesian neural networks, model selection, width-depth tradeoffs, double descent, and functional diversity in loss surfaces. The authors claim the effective dimensionality leads to a richer understanding of the interplay between parameters and functions in deep neural networks models. In their experiments the authors show that effective dimensionality compares favourably to alternative norm- and flatness- based generalization measures.

Strengths:

1 - The authors include a description of how to compute a scalable approximation to the effective dimensionality using the Lanczos algorithm and Hessian vector products.

2 - The authors include some novel experimental results showing the effective dimensionality with respect to changes in width and depth. These results are informative in how changes in depth and width affect this metric in a different way. The same for the experiments with the double descent curve.

Weaknesses:

1 - For some reason the authors seem to have taken the concept of effective dimensionality from David Mackay's approximation to the model evidence in neural networks and ignored all the extra terms in such approximation. It is currently unclear why there is a need to do this and focus only on the effective dimensionality. Almost all the experiments that the authors describe could have been done using a similar approximation to Mackay's model evidence. It is unclear why is there a need to focus just on a part of Mackay's approximation. The fact that the authors state that the effective dimensionality is only meaningful for models with low train loss seems indicative that David Mackay's approximation to the model evidence would be a better metric.

2 - With the exception of the experiments for changes in the effective dimensionality as a function of the depth and width and the double descent curve, all the other experiments and results are expected and not new to anyone familiar with David Mackay's work.

3 - The experiments on depth and width are for only one dataset and may not be representative in general. The authors should consider other additional datasets.

The authors should improve the paper, including a justification for using only the effective dimensionality and not David Mackay's approximation to the model evidence. They should also strengthen the experiments by comparing with David Mackay's approximation to the model evidence and should consider additional datasets as mentioned above.

---

> ### Author Response · Authors · 2021-02-16
> **We respectfully disagree with this assessment**
>
> We respectfully disagree with this assessment, as follows:
>
> (1) This paper represents the first time any generalization measure has been successfully used to track or provide insights into double descent, which is a *substantial* contribution. Double descent is one of the most widely visible and poorly understood phenomena in modern deep learning.
>
> (2) Similarly, this is one of the first and only works to actually consider width-depth trade-offs in neural network generalization. Most recent works focus exclusively on width. We show here how width-depth trade-offs for many sizes of convolutional networks interact for generalization, how effective dimension can be used to track these trade-offs, and how parameter counting misses a lot of relevant information in determining generalization. This is a substantial contribution.
>
> (3) We show that effective dimension is a compelling alternative to the *most successful modern generalization measures*, including PAC-Bayes flatness measures and spectral norms, selected from a thorough empirical study of many measures (Jiang et al. 2019).
>
> (4) We show how effective dimension and functional homogeneity in subspaces given by Hessian eigenvectors can be used to explain how it is surprisingly possible to dramatically compress neural networks through pruning or subspace inference. Compression is of great practical importance, and it has not been previously understood. Therefore these insights also form a substantial contribution.
>
> (5) This paper meticulously addresses parameter counting as a proxy for model complexity, which pervades the narrative in modern deep learning, and is behind many phenomena that are considered to be surprising, such as double descent. Even the popular expression "overparametrization" is an artifact of parameter counting. Many works on deep learning generalization open by expressing surprise at how models with more parameters than data points can provide good generalization. Addressing this narrative head-on is a substantial contribution.
>
> (6) All of the experiments in our paper are new contributions. The claim that an experiment is not new is not substantiated in the meta-review and is not true.
>
> (7) Effective dimensionality predates the work of MacKay (1992), and it is not tied specifically to a Laplace approximation. As we detail in the paper, effective dimensionality has a rich history, which includes the work of Cleveland (1979) and Gull (1989). Our paper is not at all about looking at one part of an approximation but not another part. Moreover, when models all have similar training loss, data fit terms are not relevant. Modern deep architectures -- the subject of this paper -- all have near-zero training loss. However, incidentally to our paper, ED does tend to be more reliable than Laplace approximations in comparing modern deep architectures, and we will highlight this point in future revisions.